# HAPLN2 forms aggregates and promotes microglial inflammation during brain aging in mice

Ayaka Watanabe[1ʘ], Shoshiro Hirayama[1ʘ], Itsuki Kominato[1], Sybille Marchese[2], Pietro Esposito[2], Vanya Metodieva[2], Taeko Kimura[3], Hiroshi Kameda[4], Terunori Sano[5], Masaki Takao[5], Sho Takatori[3], Masato Koike[4], Juan Alberto Varela[6], Taisuke Tomita[3], Shigeo Murata[1]*

1 Laboratory of Protein Metabolism, Graduate School of Pharmaceutical Sciences, The University of Tokyo, Tokyo, Japan, 2 School of Biology, University of St Andrews, St Andrews, United Kingdom, 3 Laboratory of Neuropathology and Neuroscience, Graduate School of Pharmaceutical Sciences, The University of Tokyo, Tokyo, Japan, 4 Department of Cell Biology and Neuroscience, Juntendo University Graduate School of Medicine, Tokyo, Japan, 5 Department of Clinical Laboratory, National Centre of Neurology and Psychiatry, Tokyo, Japan, 6 School of Physics and Astronomy, University of St Andrews, St Andrews, United Kingdom

ʘ These authors contributed equally to this work.
* smurata@g.ecc.u-tokyo.ac.jp

## Abstract

Protein aggregation is a hallmark of neurodegenerative diseases and is also observed in the brains of elderly individuals without such conditions, suggesting that aging drives the accumulation of protein aggregates. However, the comprehensive understanding of age-dependent protein aggregates involved in brain aging remains unclear. Here, we investigated proteins that become sarkosyl-insoluble with age and identified hyaluronan and proteoglycan link protein 2 (HAPLN2), a hyaluronic acid-binding protein of the extracellular matrix at the nodes of Ranvier, as an age-dependent aggregating protein in mouse brains. Elevated hyaluronic acid levels and impaired microglial function reduced the clearance of HAPLN2, leading to its accumulation. HAPLN2 oligomers induced microglial inflammatory responses both *in vitro* and *in vivo*. Furthermore, age-associated HAPLN2 aggregation was also observed in the human cerebellum. These findings suggest that HAPLN2 aggregation results from age-related decline in brain homeostasis and may exacerbate the brain environment by activating microglia. This study provides new insights into the mechanisms underlying cerebellar aging and highlights the role of HAPLN2 in age-associated changes in the brain.

## Introduction

Protein aggregation is associated with a wide range of physiological declines, including neurodegenerative diseases, with amyloid β (Aβ) and tau being prominent

**Data availability statement:** This study includes proteomic data obtained by mass spectrometry, which are available through the ProteomeXchange Consortium via the Japan ProteOme STandard (jPOST) partner repository (https://repository.jpostdb.org/). The dataset identifiers are JPST003565 and PXD060043 for Fig 1A, and JPST003863 and PXD064908 for Fig 1C. All other relevant data are included within the article and its Supporting Information files.

**Funding:** This work was supported by JSPS KAKENHI (grant no. JP22H00402, JP23H04918 for Sh.M., 23H04254, 22H04637 for S.H., and JP22H04926 for M.K.,) Takeda Science Foundation for Sh.M. and T.T., the European Research Council (#804581), Alzheimer's Research UK (ARUK-PPG2019A-005,) Alzheimer's Society (#AS-PhD-19a-016,) the Wellcome Trust Institutional Strategic Support Fund (204821/Z/16/Z) for J.A.V., and AMED under Grant Number JP21wm0425019 for M. T. The URLs of the funding organizations are as follows: KAKENHI: https://kaken.nii.ac.jp/en/, Takeda Science Foundation: https://www.takeda-sci.or.jp/en/, the European Research Council: https://erc.europa.eu/homepage, Alzheimer's Research UK: https://www.alzheimersresearchuk.org/, Alzheimer's Society: https://www.alzheimers.org.uk/, the Wellcome Trust Institutional Strategic Support Fund: https://wellcome.org/grant-funding/funded-people-and-projects/institutional-strategic-support-fund, AMED: https://www.amed.go.jp/en/index.html.The sponsors or funders had no role in the study design, data collection and analysis, decision to publish, or preparation of the manuscript.

**Competing interests:** The authors have declared that no competing interests exist.

examples. These proteins aggregate to cause neurodegenerative diseases such as Alzheimer's disease, disrupting neuronal function and ultimately causing cell death. Intracellular aggregates interfere with essential cellular processes such as protein folding, trafficking, and degradation, resulting in the accumulation of misfolded proteins and cellular stress [1]. This disruption impairs crucial neuronal functions, including transcription [2], mitochondrial function [3], and stress granule dynamics [4,5]. In neurodegenerative diseases, the accumulation of protein aggregates is also observed in the extracellular space. These aggregates activate microglia, triggering the release of pro-inflammatory cytokines and neurotoxic factors, leading to chronic inflammation detrimental to neuronal health [6].

The impairment of clearance systems in both intracellular and extracellular spaces is thought to be responsible for the accumulation of aberrant protein aggregates. The main intracellular pathways for clearing abnormal proteins are the ubiquitin-proteasome system [7] and the autophagy-lysosome system [8,9]. Additionally, extracellular mechanisms such as phagocytosis, secretion of proteases, and digestive exophagy by microglia [10–12], and the glymphatic system contribute to the clearance of protein aggregates from the brain [13]. However, aging is associated with a decline in the capacity of these mechanisms to maintain proteostasis, making organisms more susceptible to protein aggregation. This decline is attributed to factors such as reduced efficiency of protein degradation pathways, decreased expression of chaperone proteins, impaired cellular stress responses, and microglial senescence [14,15].

Interestingly, aggregates of wild-type Aβ, tau, and TMEM106B have been observed in the brains of non-demented elderly individuals [16–18]. This suggests that protein aggregation is not merely a consequence of neurodegenerative diseases but also an inherent aspect of the aging process itself, raising the possibility that age-dependent protein aggregates contribute to the broader physiological decline during aging.

Indeed, the detrimental effects of age-related protein aggregation have been demonstrated in model organisms. In *Caenorhabditis elegans*, the accumulation of both intracellular and extracellular protein aggregates is associated with reduced life span, regardless of disease state. In contrast, removing these aggregates extends life span. For example, deficiency in *daf-16*, a transcription factor that regulates several antioxidant and chaperone genes, or in *hsf-1*, a master transcription factor of the heat shock response, induces protein aggregation and shortens life span [19–21]. Conversely, reduced function of *daf-2*, a homolog of the mammalian insulin/insulin-like growth factor-1 receptor that suppresses *daf-16* activity, suppresses protein aggregation and extends life span [22]. Furthermore, analysis of RIPA buffer-insoluble and sodium deoxycholate-insoluble proteins by mass spectrometry has revealed that protein aggregation occurs extensively during physiological aging in the absence of disease [23,24]. These findings suggest that the clearance of age-related protein aggregates in *C. elegans* is crucial for maintaining individual health.

An extracellular protein quality control mechanism exists that disposes of extracellular waste in *C. elegans*. Coelomocytes remove extracellular debris through non-specific phagocytosis [25]. Moreover, overexpression of extracellular chaperones

in the pseudocoelom suppresses the accumulation of aggregates and extends life span [26]. These findings highlight the importance of extracellular protein quality control and emphasize the essential role of maintaining proteostasis in the extracellular space to ensure healthy aging.

In contrast to these studies about *C. elegans*, less is known about how age-dependent protein aggregates contribute to the aging process in mammals. To address this gap, we employed proteomic analyses to identify proteins that aggregate and accumulate in the mouse brain with age. Our investigation identified hyaluronan and proteoglycan link protein 2 (HAPLN2), a perinodal extracellular matrix constituent protein, as an age-dependent protein aggregate. We found that HAPLN2 accumulates in the cerebellar white matter and is mislocalized from its typical location at the nodes of Ranvier in aged mouse brains. Further investigation using recombinant protein revealed that HAPLN2 forms aggregates under acidic conditions and in the presence of hyaluronic acid, mimicking age-related changes in the brain microenvironment. Importantly, we also observed age-dependent HAPLN2 aggregation in human brain sections. These findings suggest that HAPLN2 aggregation may be a conserved hallmark of brain aging across mammalian species. Our study highlights the importance of investigating age-related protein aggregation beyond the context of specific neurodegenerative diseases and suggests a potential role for HAPLN2 aggregates in brain aging, possibly by inducing microglial activation and neuroinflammation.

## Results

### Proteomic analyses of aged mouse brains identified proteins that accumulate in the sarkosyl-insoluble fraction with age

To identify proteins that aggregate in an age-dependent manner, we enriched detergent-insoluble proteins from whole brains of young (2-month-old) and aged (24-month-old) mice using 1% N-lauryl sarcosine (sarkosyl) fractionation (Fig 1A). Sarkosyl is a well-established detergent that solubilizes natively folded proteins while enriching protein aggregates associated with neurodegenerative diseases in its insoluble P2 fraction (Fig 1A) [27–32]. Proteins in the sarkosyl-insoluble P2 fraction were subjected to label-free quantitative proteomics. We identified 57 proteins that showed more than a 5-fold increase in the brains of aged mice compared to young mice and considered them candidates for age-dependent protein aggregates (S1–S3 Tables). To elucidate the characteristics, we conducted gene ontology (GO) term enrichment analysis on these proteins (Fig 1B). The term "hyaluronic acid binding" was significantly enriched, encompassing aggrecan core protein (ACAN), hyaluronan and proteoglycan link protein (HAPLN) 1, HAPLN2, and versican core protein (VCAN), which were previously reported to increase with age in rodent brains [33–35]. In addition, tumor necrosis factor-inducible gene 6 protein (TNFAIP6), which also possesses hyaluronic acid binding ability and is known to be involved in extracellular matrix remodeling and inflammatory responses, was detected among these proteins [36].

Age-related accumulation of proteins associated with neurodegenerative diseases is attributed to their inherent resistance to degradation due to aggregation and age-related declines in clearance mechanisms [37,38]. Based on this, we hypothesized that proteins prone to aggregation in the aging mouse brain may also exhibit increased total abundance compared to those in the brains of younger mice. To determine whether these proteins also increase in total abundance, we performed further mass spectrometry analyses. Whole brains from young and aged mice were lysed with phase transfer surfactant (PTS) buffer (0.5% deoxycholic acid and 0.33% sarkosyl), to increase the solubility of hydrophobic proteins, including proteins that remain insoluble in sarkosyl alone, and to enhance their trypsin digestion [39]. The PTS-soluble supernatant, which contains most of the brain proteome (S1 Fig) and mainly corresponds to the proteins shown in S1 and partly in S2 fractions of Fig 1A, was subjected to label-free proteomic analysis (Fig 1C). We identified 28 proteins that significantly increased by more than 2-fold in the aged mouse brains compared to young mouse brains (S4–S6 Tables). Finally, we narrowed down the candidates to seven proteins showing an age-dependent increase in both proteomic analyses (Fig 1D). Although HTRA1 showed the largest fold change in Fig 1A, we could not obtain antibodies to validate this

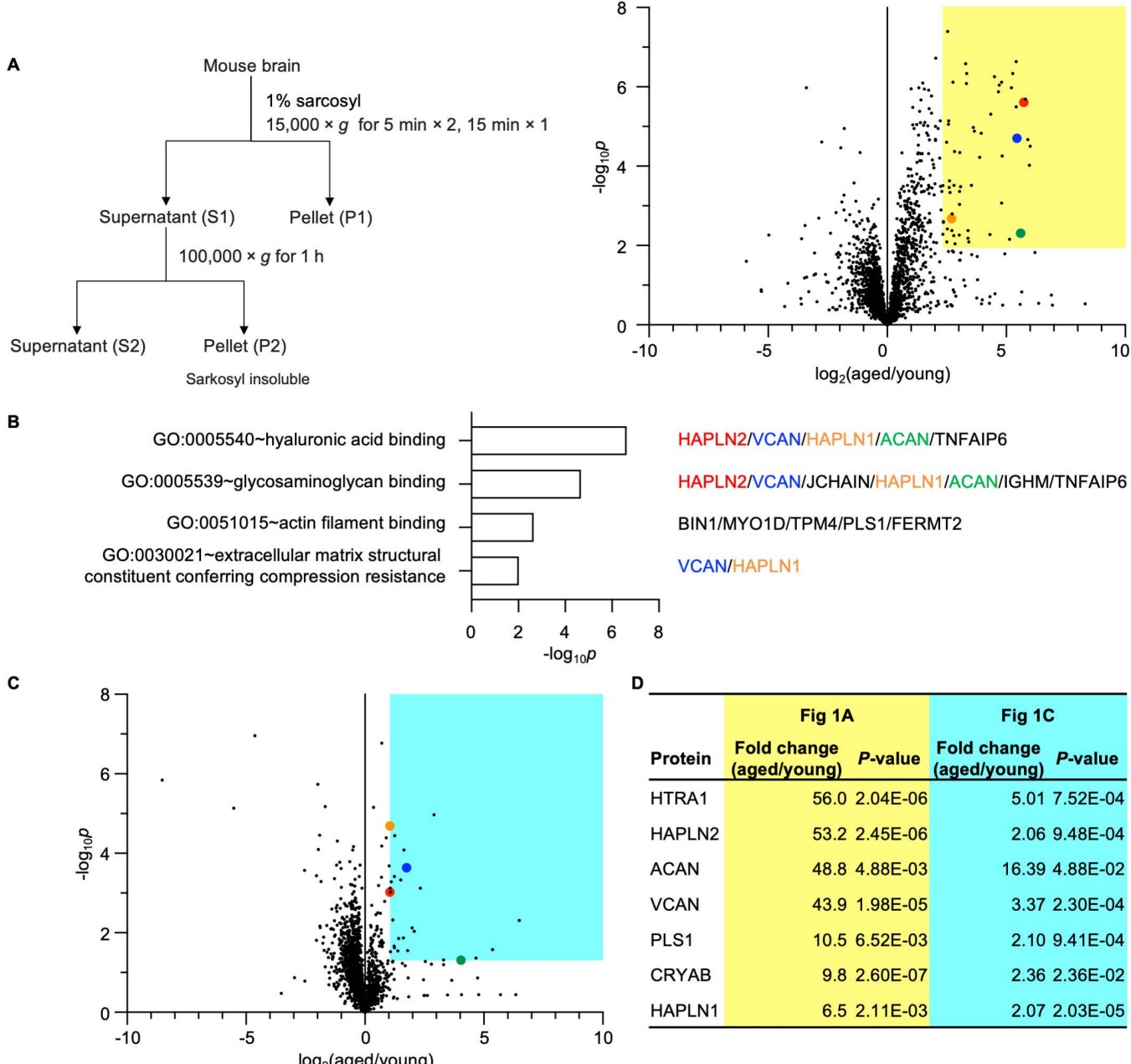

**Fig 1. Mass spectrometry analyses revealed sarkosyl-insoluble proteins accumulated in aged mouse brains.** (A) Workflow (left) illustrating the fractionation of young (2-month-old) and aged (24-month-old) mouse brains to obtain sarkosyl-insoluble protein aggregates. A volcano plot (right) represents the results of label-free quantitative proteomic analysis. The cut-off was set at a false discovery rate (FDR) of 0.01. The original results table and results with imputation of the missing values are listed in S1 and S2 Tables, respectively. The colored region in the volcano plot highlights proteins with a fold change > 5.0 and $P$-value < 0.01, as listed in S3 Table. n = 5. Specific proteins are highlighted: ACAN (green), HAPLN1 (orange), HAPLN2 (red), and VCAN (blue). (B) Bar chart showing gene ontology (GO) terms in the molecular function category enriched among the proteins identified in (A). GO enrichment analysis was performed using the Database for Annotation, Visualization, and Integrated Discovery (DAVID). Proteins included in the enriched categories are indicated on the right side of the chart. (C) Label-free proteomic analysis of phase transfer surfactant (PTS)-soluble supernatant fractions from young (2-month-old) and aged (24-month-old) mouse brains. The cut-off was set at FDR = 0.01. The original results table and results with imputation of missing values are listed in S4 and S5 Tables, respectively. Proteins with a fold change > 2.0 and $P$-value < 0.05 are highlighted in the colored regions of the volcano plots and listed in S6 Table. n = 4. The underlying data can be found in S1 Data. (D) A list of proteins significantly increased in both proteomic analyses described in (A) and (C). $P$-values by multiple unpaired $t$ test and two-stage step-up (Benjamini, Krieger, and Yekutieli) was

used to correct for multiple comparisons. Data underlying this figure are available from the ProteomeXchange Consortium via the jPOST partner repository (https://repository.jpostdb.org/) with the dataset identifiers JPST003565 and PXD060043 for (A), and JPST003863 and PXD064908 for (C).

result. Additionally, a reliable antibody against VCAN could not be obtained for use in western blotting or immunohistochemistry. Therefore, we focused on ACAN, HAPLN1, and HAPLN2 as candidates for proteins that aggregate and accumulate with age.

## HAPLN1 and HAPLN2 formed salkosyl-insoluble protein assemblies in aged mouse brains

To validate the proteomic findings, we performed immunoblotting on each fraction prepared as shown in Fig 1A. Consistent with the proteomic analyses, marked age-dependent increases in ACAN, HAPLN1, and HAPLN2 were observed in the sarkosyl-insoluble P2 fraction (Figs 2A and 2B). As expected from previous studies, ubiquitinated proteins also increased in the P2 fraction with age (Figs 2A and 2B), confirming the successful enrichment of protein aggregates [33]. These results also demonstrated the sarkosyl-insolubility of ACAN, HAPLN1, and HAPLN2 in aged mouse brains.

Characteristics of protein aggregates include not only detergent-insolubility but also the formation of high-molecular-weight assemblies trapped on nitrocellulose membranes and aberrant localization as large deposits *in vivo* [34,35]. Therefore, we performed a series of experiments to determine whether ACAN, HAPLN1, and HAPLN2 are aggregated in aged mouse brains.

To investigate whether ACAN, HAPLN1, and HAPLN2 form sarkosyl-insoluble large protein assemblies, we performed a filter trap assay using the S1 fraction. To simultaneously detect large protein assemblies and total protein, we used a dual membrane filter trap assay by sequentially layering a nitrocellulose membrane (0.45-μm pore size) on top of a PVDF membrane (0.2-μm pore size) [34,40,41]. The nitrocellulose membrane captures large protein assemblies, while proteins passing through are trapped by the PVDF membrane, which has a higher protein-binding capacity. Our results showed that HAPLN1 and HAPLN2 trapped on the nitrocellulose membrane increased markedly with age, whereas ACAN did not differ between young and aged mice (Fig 2C). In contrast, all three proteins increased on the PVDF membrane in aged mice. These findings indicate that HAPLN1 and HAPLN2 form sarkosyl-insoluble large complexes with age, while ACAN does not. ACAN, HAPLN1, and HAPLN2 are known to form the extracellular matrix via non-covalent association with hyaluronic acid. Previous studies showed that ACAN from aged mouse brain is enriched in the Triton X-100-insoluble fraction but becomes soluble after degrading high-molecular-weight hyaluronic acid [42]. Therefore, we next investigated whether treatment with hyaluronidase solubilizes ACAN, HAPLN1, and HAPLN2 from the sarkosyl-insoluble P2 fraction. We treated the sarkosyl-insoluble P2 fraction with hyaluronidase, followed by ultracentrifugation to yield the supernatant hyaluronidase-extractable S3 and the pellet hyaluronidase-resistant P3 fractions (Fig 2D). ACAN was recovered in the S3 fraction after hyaluronidase treatment (Fig 2E). In contrast, the majority of HAPLN1 and HAPLN2 remained in the P3 fraction, with only a portion shifting to the S3 fraction (Fig 2E). We also performed a filter trap assay on the S3 and P3 fractions. HAPLN1 and HAPLN2 in the P3 fraction were captured on a nitrocellulose membrane, whereas ACAN in the P3 fraction was not captured on a nitrocellulose membrane (Fig 2F). These results suggest that HAPLN1 and HAPLN2 form high molecular weight protein assemblies that remain insoluble in sarkosyl despite hyaluronic acid degradation. We focused on HAPLN2 for further investigation because HAPLN2 showed the highest fold change in western blot analysis among ACAN, HAPLN1, and HAPLN2, and its aggregates were not solubilized by hyaluronidase treatment (Fig 1D).

## HAPLN2 was deposited in the extracellular space in the cerebellar white matter of aged mice

To investigate whether HAPLN2 forms aberrant deposits in aged mouse brains, we examined the distribution and morphological structure of HAPLN2 in mouse brains. Previous studies have shown that HAPLN2 is expressed in the cerebellar white matter of young adult mice [43,44]. Immunoblot analysis confirmed that HAPLN2 protein was

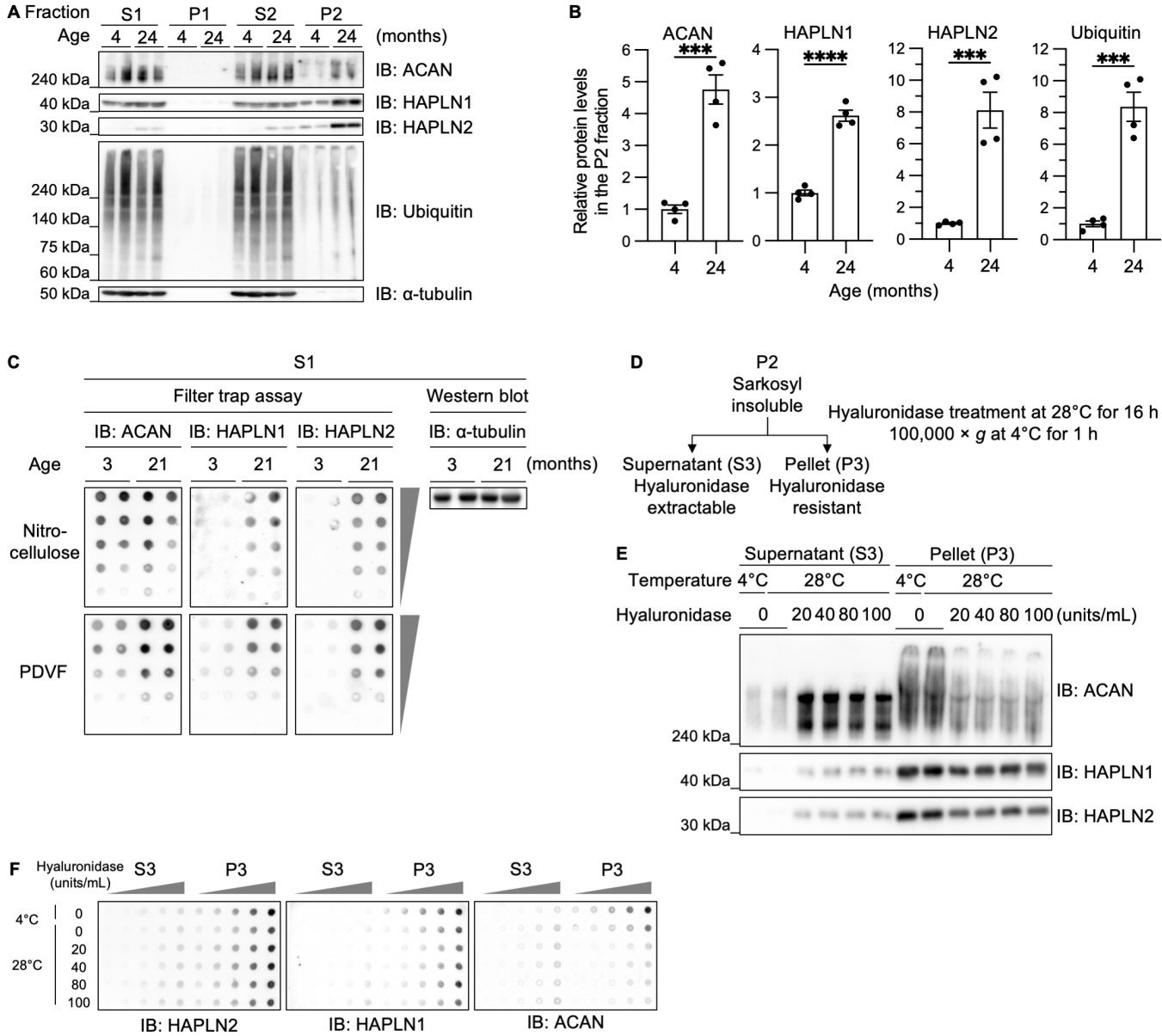

**Fig 2. Extracellular matrix component proteins accumulate in the sarkosyl-insoluble fraction of mouse brains with age.** (A) Immunoblot analyses of ACAN, HAPLN1, HAPLN2, and ubiquitin in the S1, P1, S2, and P2 fractions, as outlined in Fig 1A, from mouse brains. n = 2. (B) Densitometric quantitation of (A). The expression levels of the indicated proteins were normalized to α-tubulin in the S1 fractions. Error bars represent mean ± S.E.M. P-values by two-tailed Student t test. ***$p < 0.001$, ****$p < 0.0001$. n = 4. The underlying data can be found in S1 Data. (C) Filter trap assay and immunoblot analysis of the S1 fraction from young (3-month-old) and aged (21-month-old) mouse brains. n = 2. (D) Diagram illustrating hyaluronidase treatment of the sarkosyl-insoluble P2 fraction. The P2 fraction was treated with hyaluronidase and centrifuged at 100,000 g. (E) Immunoblot analyses of ACAN, HAPLN1, and HAPLN2 in the hyaluronidase-extractable S3 and hyaluronidase-resistant P3 fractions from 27-month-old mouse brains. n = 3. (F) Filter trap assay of the fractions described in (E). n = 3.

predominantly detected in the cerebellum of 24-month-old mice, whereas it was only faintly detected in the cerebellum of 4-month-old mice (Fig 3A). Immunohistochemical analysis revealed that the expression of HAPLN2 increased with age in highly myelinated regions, such as brain stem, corpus callosum, and the cerebellar white matter, which is mainly

composed of myelinated axons expressing myelin basic protein (MBP) and is surrounded by granule layers stained with 4′,6-diamidino-2-phenylindole (DAPI) (Figs 3B and S2A). These results demonstrate that HAPLN2 accumulates in highly myelinated regions of aged mice.

We next examined its detailed structure in the cerebellar white matter of aged mice, where the age-dependent accumulation of HAPLN2 was most prominent. Immunohistochemistry analysis showed significantly enlarged puncta of HAPLN2 in the aged mouse brains compared to young mouse brains (Figs 3C and 3D). We then investigated whether the HAPLN2 deposits are localized to the intracellular or extracellular space. Hyaluronic acid is a major component of the extracellular matrix, which is visualized by staining with hyaluronic acid binding protein (HABP). In young mouse samples, HABP staining was weak, and colocalization of HAPLN2 and HABP was unclear. In contrast, in aged mouse brains, both HABP and HAPLN2 staining showed significantly enlarged puncta, and colocalization of HAPLN2 and HABP was clearly observed (Figs 3E–G, and S2B). These results indicate that HAPLN2 accumulates as large deposits with hyaluronic acid in the extracellular space of the cerebellar white matter of aged mice.

To determine which cells in the cerebellum express HAPLN2 and whether the accumulation of HAPLN2 is due to increased mRNA expression or not, we referred to the results of a previous single-cell transcriptome analysis. This data revealed that *Hapln2* mRNA expression was specifically detected in oligodendrocytes and that mRNA levels remained constant between 2–3 months and 21–22 months of age (Fig 3H) [45]. These results suggest that the age-dependent accumulation of HAPLN2 protein is not due to the induction of mRNA expression.

Since the single-cell RNA-seq result indicates that HAPLN2 is expressed in oligodendrocytes, we performed co-staining of HAPLN2 and the oligodendrocyte marker MBP [46]. However, the HAPLN2 deposits did not colocalize with oligodendrocytes (Fig 3I). We also examined co-staining of HAPLN2 with ubiquitin and p62, as protein aggregates that initially form intracellularly and are later deposited extracellularly, as is typically observed with tau, often colocalize with these proteins [47]. However, HAPLN2 did not colocalize with ubiquitin and p62 (S2C and S2D Fig). These results suggest that HAPLN2 is secreted from oligodendrocytes and subsequently accumulated in the extracellular space.

We next investigated whether enlarged HAPLN2 puncta retain their physiological function. HAPLN2 is normally localized to the perinodal extracellular matrix, which surrounds the nodes of Ranvier and contains hyaluronic acid [48]. Therefore, we examined the colocalization between HAPLN2 and contactin-associated protein (Caspr), which localizes to the paranodal junction. The colocalization of HAPLN2 with Caspr decreased with age (Figs 3J and 3K). In addition, Mander's overlap coefficient (MOC) M1, which indexes the proportion of HAPLN2 colocalized with Caspr, decreased with age (Fig 3L), suggesting that HAPLN2 was abnormally localized outside the nodes of Ranvier in aged mice. In contrast, MOC M2, an index of the proportion of Caspr colocalized with HAPLN2, did not change with age (Fig 3M), suggesting that most of Caspr colocalizes with functional HAPLN2 and that the nodes of Ranvier are not dysregulated in aged mice. Additionally, the area of HAPLN2 mislocalized from Caspr was larger than HAPLN2 colocalized with Caspr in 24-month-old mice (Fig 3N). This further supports that enlarged HAPLN2 puncta have lost their function.

To investigate whether the accumulation of HAPLN2 is associated with age-related demyelination, brain slices from young (3-month-old) and aged (18-month-old) mice were stained for HAPLN2, calbindin 1, a marker of Purkinje cells, the most abundant neuronal type in the cerebellum, and MBP. Immunohistochemical analysis showed that, with aging, MBP-negative regions increasingly appeared as hole-like structures within the white matter, and these regions exhibited a corresponding increase in calbindin 1 staining, as indicated by the magenta arrows in the inset figures (S3A and S3B Figs). This pattern was consistent with the demyelination-associated changes observed by Luxol Fast Blue (LFB) staining [49,50]. Additionally, the calbindin 1-positive area was increased in aged mice compared to young mice, suggesting that demyelination exposes the axons of Purkinje cells (S3C Fig). Specifically, while the calbindin 1-positive area accounted for approximately 1.2% in young mice, it increased nearly 3-fold in aged mice to approximately 3.3%.

To directly assess the relationship between demyelination severity and HAPLN2 accumulation, we compared cerebellar regions in aged mice exhibiting high versus low calbindin 1 staining (S3D and S3E Fig). Specifically, calbindin 1-high and

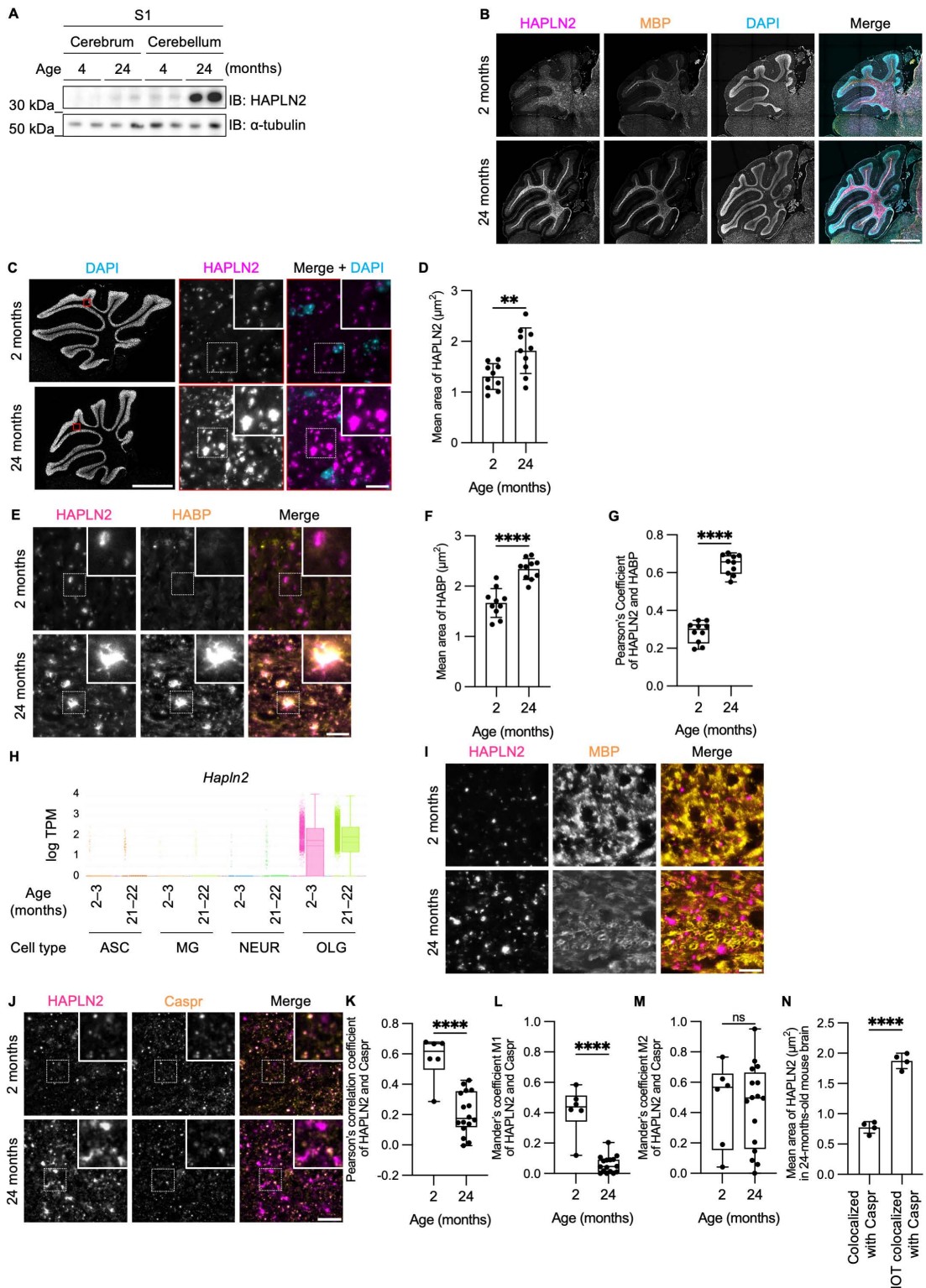

**Fig 3. HAPLN2 forms aggregate-like structures in the cerebellar white matter of aged mice.** (A) Immunoblot analysis of the S1 fraction from the cerebrum and cerebellum of mice. n = 2. (B) Representative images of fluorescence immunohistochemistry of the mouse cerebellum showing HAPLN2, myelin basic protein (MBP, a marker for myelinated axons), and 4′,6-diamidino-2-phenylindole (DAPI). n = 3. (C) Magnified fluorescence

immunohistochemistry images for HAPLN2 corresponding to the boxed areas in the left panels. n = 3. (D) Quantitation of the mean area of HAPLN2-positive puncta in (C). (E) Representative images of fluorescence immunohistochemistry for HAPLN2 and biotinylated hyaluronic acid-binding protein (HABP), a probe for hyaluronic acid. n = 3. (F) Quantitation of the mean area of HABP-positive puncta in (E). (G) Pearson's correlation coefficients for colocalization between HAPLN2 and HABP. (H) Single-cell transcriptomics data from [45]. TPM: transcripts per million. ASC: astrocyte; MG: microglia; NEUR: mature neuron; OLG: oligodendrocyte. Error bars indicate maximum values, boxes indicate the first and third quartiles, and solid and dashed lines represent the median and mean, respectively. (I) Representative images of fluorescence immunohistochemistry for HAPLN2 and MBP as an oligodendrocyte marker in the cerebellar white matter. n = 3. (J) Representative images of fluorescence immunohistochemistry for HAPLN2 and Caspr, a marker for the nodes of Ranvier. n = 3. (K–M) Colocalization analysis between HAPLN2 and Caspr using Pearson's correlation coefficient (K) and Mander's overlap coefficients (MOC) M1 (L) and M2 (M). MOC M1 represents the proportion of HAPLN2 overlapping Caspr, while MOC M2 represents the reverse. (N) Quantitation of the mean area of HAPLN2 punctum, categorized by whether it colocalizes with Caspr in 24-month-old mice. All coefficient calculations were performed using ImageJ (version 1.53t) with the JACoP plugin. Scale bars: 1 mm (B and C (left)) and 10 μm (C (right); E; I; J). Error bars represent mean ± S.D. *P*-values by two-tailed Student *t* test. **$p < 0.01$, ***$p < 0.001$, ****$p < 0.0001$, ns = not significant. The underlying data for (D), (F), (G), (K), (L), (M), and (N) can be found in S1 Data.

calbindin 1-low regions were defined as square areas measuring 50 μm × 50 μm in which the calbindin 1-positive area was more than 5% or less than 1%, respectively. These thresholds were determined based on the quantification shown in S3C Fig, where the average calbindin 1-positive area in 18-month-old mice was 3.3%, nearly 3-fold higher than the 1.2% observed in 3-month-old mice. Quantitative analysis revealed no significant differences in HAPLN2-positive puncta size or density between these regions. These results demonstrate that HAPLN2 accumulation occurs independently of age-related demyelination severity.

Furthermore, we investigated whether other proteins are deposited in the aged brain. Immunostaining of brain slices from young and aged mice for HAPLN1 and ACAN revealed that, in aged mice, HAPLN1 formed puncta that were mislocalized relative to ACAN staining in the cerebellar nuclei, as indicated by the magenta arrows (S4A and S4B Fig). In addition, the area of HAPLN1 not colocalized with ACAN increased with age, whereas the ACAN-positive areas showed no significant age-related changes (S4C Fig). These findings suggest that HAPLN1 undergoes age-related deposition and loses its function in the perineuronal net in the aged brain.

### Deposition of HAPLN2 aggregates starts in middle age in mouse brains

Research in humans and rodents has highlighted middle age as a period marked by the onset of changes associated with brain aging [51]. To investigate when HAPLN2 starts to accumulate, we examined the age-related progression of HAPLN2 accumulation in the mouse brain. We performed sarkosyl fractionation followed by immunoblot analysis using young (3- and 6-month-old), middle-aged (12-month-old), and aged (18-month-old) mice. HAPLN2 began to accumulate in the S1 and P2 fractions from 12-month-old mice and was further enriched in 18-month-old mice (Fig 4A). Correspondingly, the puncta of HAPLN2 became larger in 12- and 18-month-old mice, consistent with the immunoblot analysis (Figs 4B and 4C). These results indicate that HAPLN2 accumulates gradually from middle age to old age, further confirming the age-dependent accumulation of HAPLN2 aggregates.

### Low pH, high NaCl concentration, and hyaluronic acid promoted HAPLN2 aggregation

To investigate the factors affecting HAPLN2 aggregation, we explored the aggregation propensity of recombinant HAPLN2 protein. Various factors have been reported to influence the formation of protein aggregates, including pH and NaCl concentration. For instance, the TAR DNA-binding protein of 43 kDa (TDP-43) aggregates at high pH and NaCl concentration [52], while fused in sarcoma (FUS) aggregates at low concentrations of RNA [53] and in the presence of dextran [54,55] and NaCl [56]. Therefore, we incubated recombinant HAPLN2 protein in buffers with varying pH and NaCl concentration, and in the presence or absence of hyaluronic acid. The formation of amorphous aggregates of HAPLN2 was promoted at low pH, high NaCl concentration, and in the presence of hyaluronic acid (Figs 5A, 5B, S5A, and S5B). Quantitative area measurements revealed that recombinant HAPLN2 formed amorphous aggregates larger than 10 μm² at pH 6.0, while it

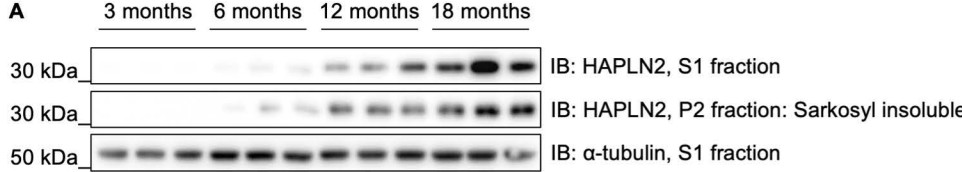

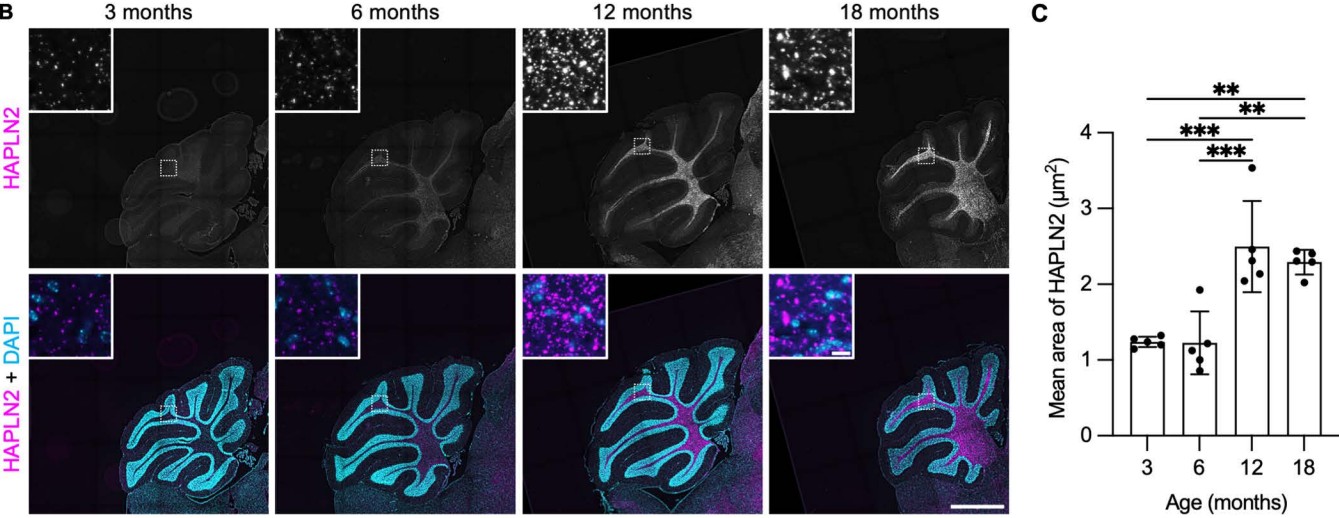

**Fig 4. Time-course analysis of age-dependent HAPLN2 accumulation in mouse brains.** (A) Immunoblot analyses of the S1 and P2 fractions from the mouse cerebellum at different ages. n = 3. (B) Representative images of fluorescence immunohistochemistry of the mouse cerebellum showing HAPLN2 and DAPI staining. n = 3. Scale bars: 1 mm (B) and 10 μm (B, inlet). (C) Quantitation of the mean area of HAPLN2-positive structures shown in (B). Error bars represent mean ± S.D. *P*-values by one-way ANOVA followed by Tukey's *post hoc* test. \*\**p* < 0.01, \*\*\**p* < 0.001. The underlying data can be found in S1 Data.

formed small oligomer-like aggregates less than 10 μm² at pH 7.0, and no aggregates were formed at pH 8.0 in 150 mM NaCl (S5C Fig). Transmission electron microscopy analysis further confirmed that HAPLN2 incubated at pH 6.0 in 150 mM NaCl formed amorphous aggregates (S5D Fig).

Many protein aggregates contain cross-β-sheet structures and form protein fibrils. However, it is impossible to distinguish between oligomeric and fibrillar structures using diffraction-limited optics [57]. To examine whether HAPLN2 aggregates contain β-sheet-mediated aggregates that are stained with thioflavin T or Nile Red, we conducted super-resolution imaging analysis (S5E Fig) [58]. In recombinant HAPLN2 incubated at pH 6.0, some spots, but not the entire aggregate, were stained with thioflavin T and Nile Red, indicating that the amorphous-appearing HAPLN2 aggregates contain cross-β-sheet structures (Fig 5C).

To analyze whether the HAPLN2 aggregates formed fibrillar structures, which are characterized by a long axis and low circularity ratio, we statistically analyzed the morphology of the Nile Red-positive clustered aggregates, according to a previous report [59]. The long axis of the Nile Red-positive clusters increased under acidic conditions (Fig 5D), but the circularity ratio did not change with pH (Fig 5E).

To determine whether aggregation of HAPLN2 induced by low pH results in an increase in β-sheet structures, we analyzed the circular dichroism (CD) spectra of the protein in both the soluble state and the aggregated state. Upon protein aggregation and transition to β-sheet conformation, a red shift of the positive band is observed in the CD spectrum. [60].

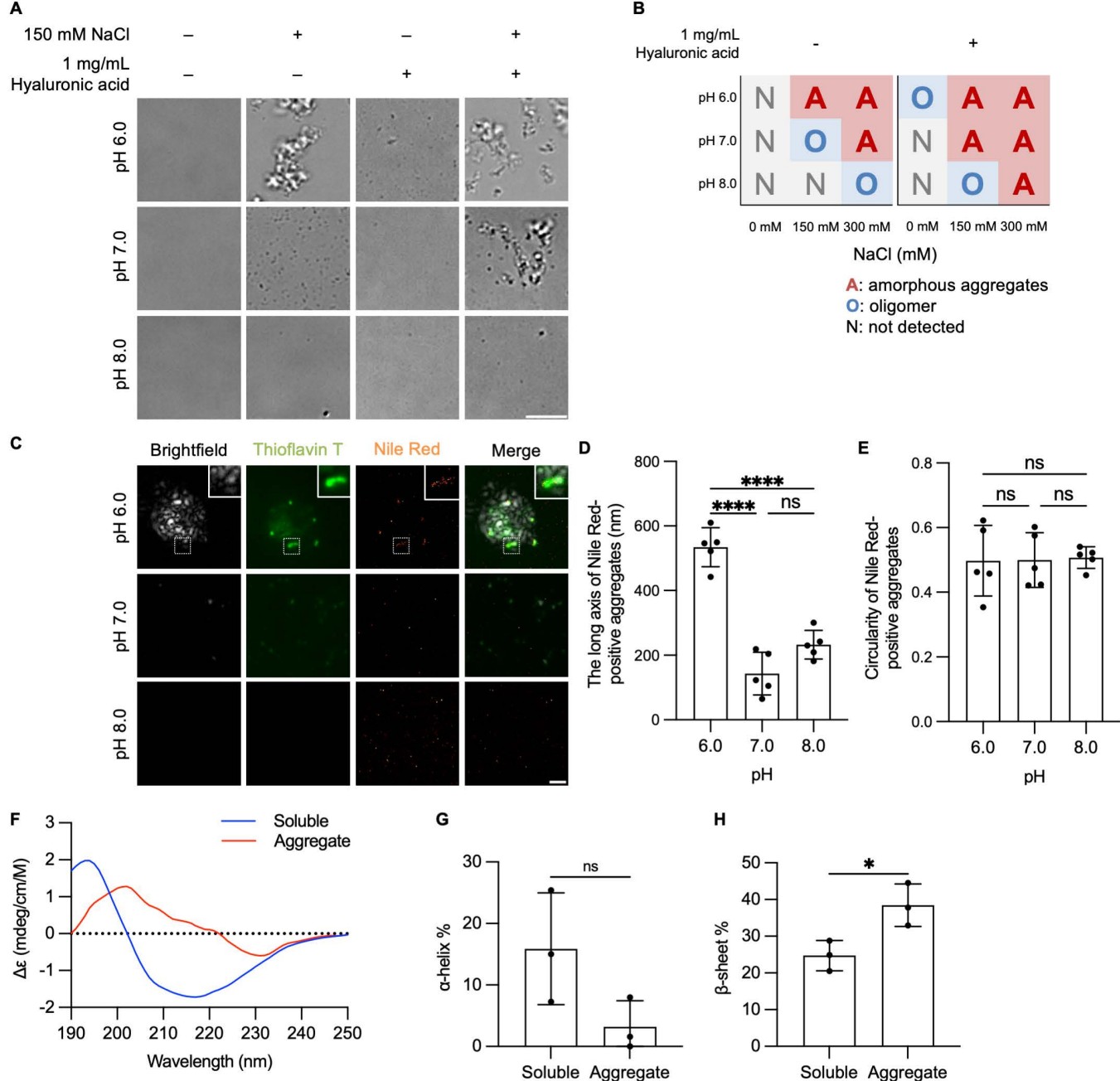

**Fig 5. Aggregates of recombinant HAPLN2 formed under low pH, high NaCl, or hyaluronic acid conditions partially contain cross-β-sheet structures.** (A) Brightfield microscopic images of recombinant HAPLN2 aggregates incubated in the indicated buffer conditions without agitation. (B) Phase diagrams showing the aggregation states of recombinant HAPLN2 under the indicated buffer conditions. (C) Brightfield and highly inclined and laminated optical sheet (HiLo) images of recombinant HAPLN2 aggregates incubated in a buffer at the indicated pH with 150 mM NaCl. Thioflavin T and Nile Red were used at concentrations of 5 μM and 5 nM, respectively to visualize cross-β-sheet structures. (D, E) Quantitative analyses of the long axis (D) and circularity ratio (E) of Nile Red-positive aggregates shown in (C). Data were collected from at least five fields of view. 500 nM Recombinant HAPLN2 monomers were incubated for 24 hours at 37ºC in all conditions. Error bars represent mean ± S.D. *P*-values by one-way ANOVA followed by Tukey's *post hoc* tests. Scale bars: 10 μm (A) and 1 μm (C). (F) CD spectra of soluble (blue) and aggregated (red) recombinant HAPLN2 protein. (G, H) Secondary structure estimation of α-helix (G) and β-sheet (H) content based on BeStSel analysis of CD spectra of (F). n = 3. Error bars represent the mean ± S.D. *P*-values were calculated using a two-tailed Student *t* test. *$p < 0.05$, ****$p < 0.0001$, ns = not significant. The underlying data for (D), (E), (F), (G), and (H) can be found in S1 Data.

We found that recombinant HAPLN2 proteins that aggregated at low pH exhibited a shift in positive ellipticity from approximately 190 nm to 200 nm, suggesting an increase in β-sheets (Fig 5F). Secondary structure analysis of the CD spectra using BeStSel further confirmed that the proportion of β-sheet structures was significantly increased in the aggregated state compared to the soluble state, while the α-helix content was reduced (Figs 5G and 5H) [61].

These results suggest that low pH induces the formation of amorphous, rather than fibrillar, aggregates of HAPLN2 containing partial cross-β-sheet structures.

### Treatment with hyaluronidase promoted the clearance of HAPLN2 aggregates in the mouse cerebellar white matter

It has been reported that the volume of the extracellular spaces reduces with age, which is thought to lead to impairment of protein aggregate clearance [62,63]. Hyaluronic acid is the major component of the perinodal extracellular matrix and accumulates in the brain with aging [64–68]. Furthermore, it has been shown that hyaluronidase treatment increases the volume of the extracellular space in mouse hippocampal slices [69]. Therefore, we hypothesized that reducing hyaluronic acid would increase the extracellular space and improve access to HAPLN2 aggregates, thereby enhancing their clearance.

To test this idea, we injected hyaluronidase into the mouse cerebellum (Fig 6A). Immunohistochemical analysis revealed that the expression and size of HAPLN2 dramatically decreased 16 hours after hyaluronidase injection (Figs 6B, 6C). Immunoblot analysis of brain lysates fractionated as in Fig 1A showed that HAPLN2 and HAPLN1 in the sarkosyl-insoluble P2 fractions in the hyaluronidase-injected brain (ipsilateral) significantly decreased compared to those in the PBS-injected ipsilateral brain, whereas HAPLN2 levels in the sarkosyl-soluble S2 fractions were not significantly affected (Figs 6D, 6E, S6A, and S6B).

Since hyaluronidase administration resulted in a marked reduction of HAPLN2 aggregates, we next investigated the underlying mechanism. A recent study has shown that hyaluronidase injection increased the volume and decreased the viscosity of the extracellular space, thereby potentially facilitating glymphatic flow [69]. Therefore, we analyzed changes in the levels of HAPLN2 in the cerebrospinal fluid (CSF) in the presence and absence of hyaluronidase treatment. Capillary western blot analysis demonstrated that HAPLN2 protein levels in the CSF were substantially increased three hours after hyaluronidase administration (Fig 6F, 6G). This observation suggests that HAPLN2 aggregates may be cleared from the white matter via pathways such as glymphatic flow, although the involvement of other clearance or degradation mechanisms cannot be completely excluded. Collectively, these results indicate that hyaluronidase treatment facilitates the removal of HAPLN2 aggregates from the cerebellar white matter.

Aging is also associated with a decline in motor function, which is attributed to cerebellar functional decline [70]. Therefore, to investigate the physiological effects of hyaluronidase treatment, we analyzed the motor function of 12-month-old mice three months after hyaluronidase injection (Fig 6H–P).An open field test was conducted for five minutes, during which the following parameters were measured: total locomotor activity indicators such as walking distance and mean velocity; indicators of burst activity including maximum velocity; proportions of time spent in different locomotor states represented by time spent moving and time spent immobile; anxiety-related spatial preference assessed by time spent incorners; and parameters related to turning behavior and balance, including mean meander and mean turn angle. The open-field test analysis revealed that hyaluronidase injection significantly increased walking distance, mean velocity, and time spent moving, while it decreased time spent immobile, mean meander, and mean turn angle, suggesting improvements in gait and balance. In contrast, maximum velocity and time spent in corners were not significantly affected, indicating that momentary motor abilities, anxiety levels, and exploratory behavior remained unchanged. These results suggest that age-dependent protein aggregates, including HAPLN2, are responsible for the decline in the health of the brain.

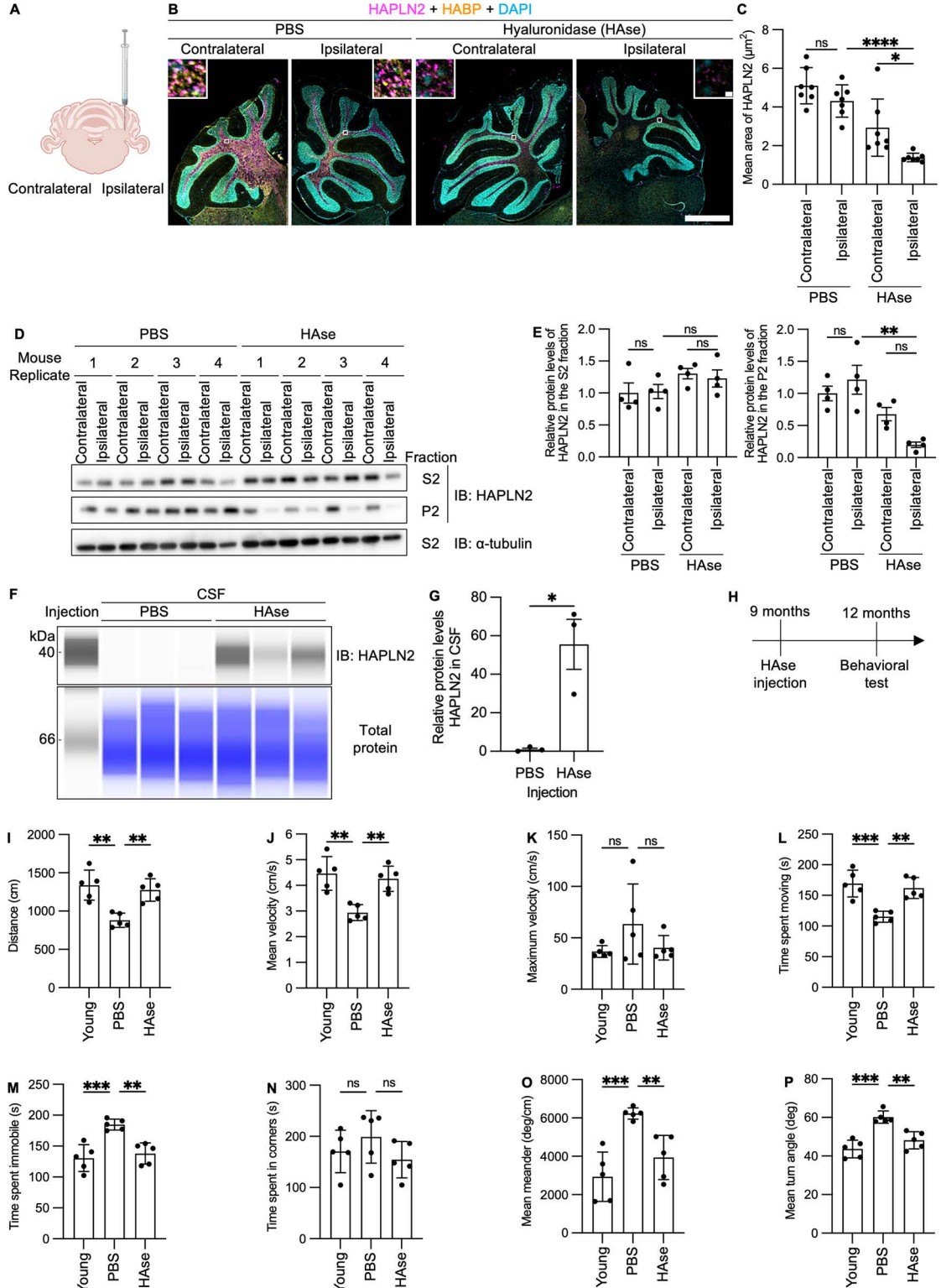

**Fig 6. Hyaluronidase promotes clearance of HAPLN2 aggregates in the mouse cerebellum.** (A) Schematic illustration of hyaluronidase injection into the mouse cerebellum. Created with BioRender.com. (B) Fluorescence immunohistochemistry images of the cerebellum of 12-month-old mice showing HAPLN2, HABP, and DAPI signals. n = 4. Scale bar: 1 mm (B), 10 μm (B, inlet). (C) Quantitation of the mean area of HAPLN2-positive puncta in (B). (D) Immunoblot analysis of the cerebellum corresponding to the samples in (B). (E) Densitometric quantification of (D). HAPLN2 Protein levels in

the S2 fraction were normalized to α-tubulin in the S2 fractions. HAPLN2 protein levels in the P2 fraction were normalized to α-tubulin in the S1 fractions. (F) Capillary western blot analysis of cerebrospinal fluid in (B). (G) Densitometric quantification of (F). HAPLN2 protein levels in CSF were normalized to total protein. (H) Schematic illustration of the time course of the hyaluronidase injection followed by the open field test. (I–P) Open field test analysis of the hyaluronidase-injected mice. The assessed parameters included: distance (I), mean velocity (J), maximum velocity (K), time spent moving (L), time spent immobile (M), time spent in corners (N), mean meander (O), and mean turn angle (P). The duration of the measurement was five minutes. Four mice were analysed in each trial. n = 5. Error bars represent mean ± S.D. (C) and S.E.M. (E, G, I–P) *P*-values by one-way ANOVA followed by Tukey's *post hoc* tests. *$p < 0.05$, **$p < 0.01$, ***$p < 0.001$, ****$p < 0.0001$, ns = not significant. The underlying data for (C), (E), (G), (I), (J), (K), (L), (M), (N), (O), and (P) can be found in S1 Data.

## Microglia regulated the accumulation of HAPLN2 aggregates

Microglia are the resident macrophages in the brain, acting as scavengers that remove debris and protein aggregates from the extracellular space through phagocytosis [71]. To examine whether microglia are involved in the clearance of HAPLN2 aggregates, we treated 4-week-old mice with pexidartinib (PLX3397), a colony-stimulating factor 1 receptor (CSF1-R) inhibitor known to deplete microglia [72]. Immunostaining with ionized calcium-binding adapter molecule 1 (Iba1), a microglia/macrophage marker, confirmed that treatment with PLX3397 for two months successfully depleted microglia (Fig 7A). Immunohistochemical analysis showed that microglial depletion resulted in the accumulation and enlargement of HAPLN2 puncta in the brains of PLX3397-treated mice (Fig 7B and 7C). Furthermore, PLX3397 treatment resulted in HAPLN2 enrichment within the sarkosyl-insoluble fraction of the cerebellum (Fig 7D and 7E). These results suggest that microglia play a critical role in the clearance of HAPLN2 aggregates.

To determine whether microglial depletion induces myelin damage and subsequently contributes to HAPLN2 aggregate accumulation, we compared brain sections from mice treated with PLX3397 to deplete microglia and from untreated controls. Immunostaining was performed using antibodies against Iba1, a microglial marker; calbindin 1, a marker of Purkinje cells; and MBP, a myelin marker. Quantitative analysis revealed no significant changes in calbindin 1-positive areas following microglial depletion, indicating that the depletion protocol used here did not affect the extent of myelin damage or Purkinje cell axon exposure. However, microglial depletion significantly reduced HAPLN2 aggregate clearance. These results suggest that microglia contribute to the clearance of HAPLN2 aggregates through mechanisms distinct from their canonical phagocytic pathway for damaged myelin (S7A–C Fig).

## Oligomers and aggregates of HAPLN2 induced microglial activation *in vitro* and *in vivo*

Microglia that phagocytose extracellular protein aggregates exhibit a pro-inflammatory M1 phenotype and secrete cytokines such as interleukin (IL)-6 and tumor necrosis factor (TNF)-α [71]. Recent studies have shown that soluble oligomers of Aβ, tau, and α-synuclein in the extracellular spaces induce inflammatory responses prior to the accumulation of protein aggregates in the brains of patients with neurodegenerative diseases [73].

To investigate whether HAPLN2 triggers microglial inflammation, we purified recombinant proteins of full-length HAPLN2 and each domain from a lipopolysaccharide (LPS)-free *Escherichia coli* strain, ClearColi BL21(DE3) (Fig 8A) [74]. First, we performed a filter trap assay to determine whether solubilized full-length HAPLN2 and the N-terminal Ig-like domain exist as monomers or soluble oligomers. Both soluble full-length HAPLN2 and the N-terminal Ig-like domain were trapped on the nitrocellulose membrane after ultracentrifugation, indicating that they form soluble oligomers (S8A Fig). We also prepared aggregated full-length HAPLN2 and the N-terminal Ig-like domain by incubating each soluble protein at pH 6.0, which was confirmed by filter trap assay (S8A Fig).

To evaluate the potential to induce microglial inflammatory responses, we added full-length HAPLN2 oligomers and aggregates, as well as the positive control LPS, to the MG6 microglial cell line. Both soluble oligomers and aggregates of HAPLN2 increased the mRNA expression of IL-6 and TNF-α (Fig 8B). These results suggest that HAPLN2 induces pro-inflammatory microglial activation in both the soluble oligomeric and aggregated states.

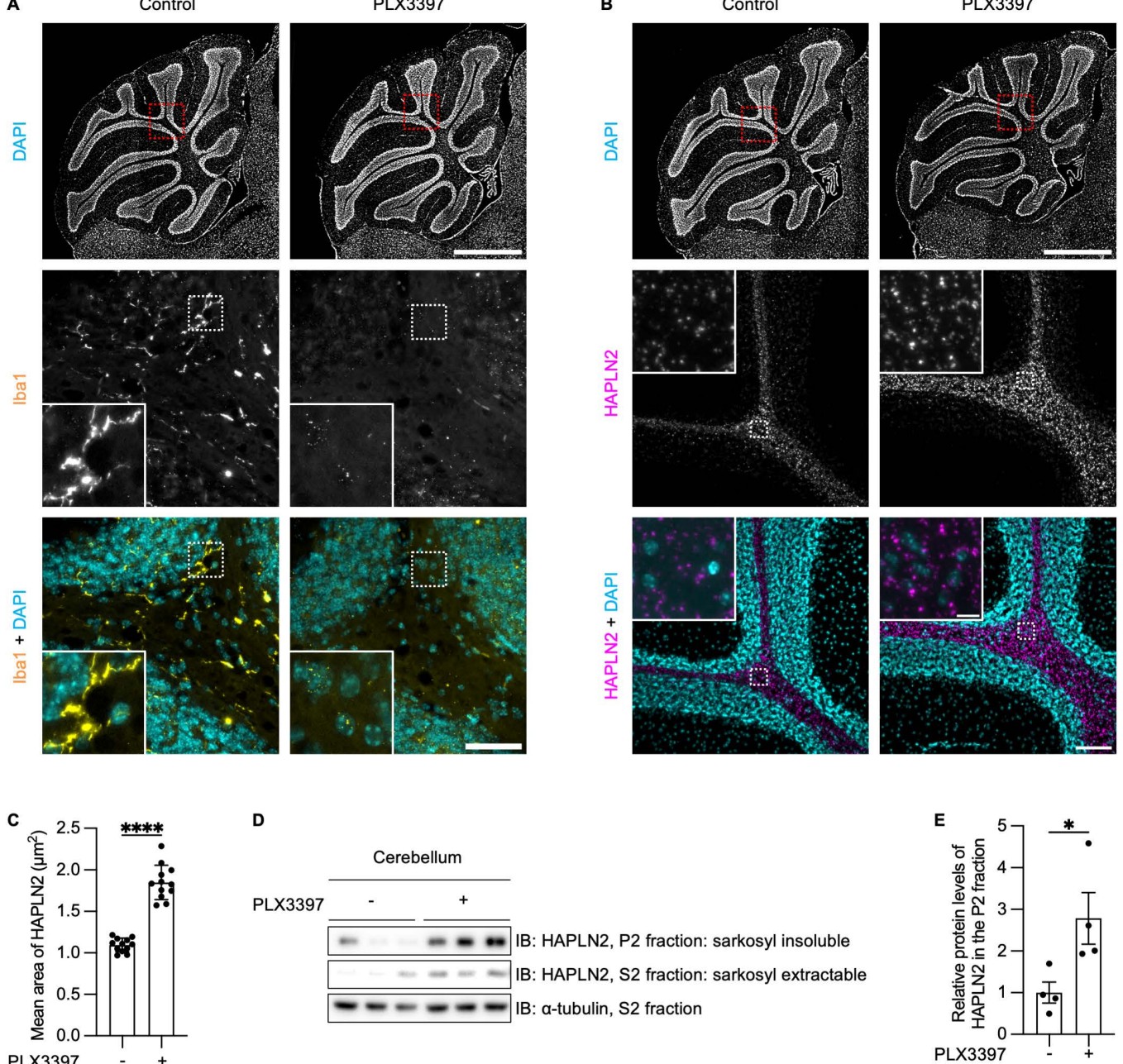

**Fig 7. Accumulation of HAPLN2 aggregates is promoted by microglial depletion in the cerebellum.** (A) Immunohistochemistry images of the cerebellum from mice fed AIN-76A control chow or AIN-76A containing 290 mg/kg PLX3397 from one month to three months of age, stained with anti-ionized calcium-binding adaptor molecule 1 (Iba1) antibody as a microglia marker. n = 3. (B) Immunohistochemistry images of the cerebellum from the mice in (A), stained with anti-HAPLN2 antibody. n = 3. (C) Quantification of the mean area of HAPLN2-positive puncta in (B). (D) Immunoblot analysis of the cerebellum from the mice in (B), fractionated as described in Fig 1A. n = 3. (E) Densitometric quantitation of (D). Protein expression levels were normalized to α-tubulin in the S1 fractions. Scale bars: 1 mm (A, up; B, up), 50 μm (A, down), and 100 μm (B, down), 10 μm (B, down, inlet). Error bars represent mean ± S.D. *P*-values were calculated using two-tailed Student *t* test. *$p < 0.05$, ****$p < 0.0001$. The underlying data for (C) and (E) can be found in S1 Data.

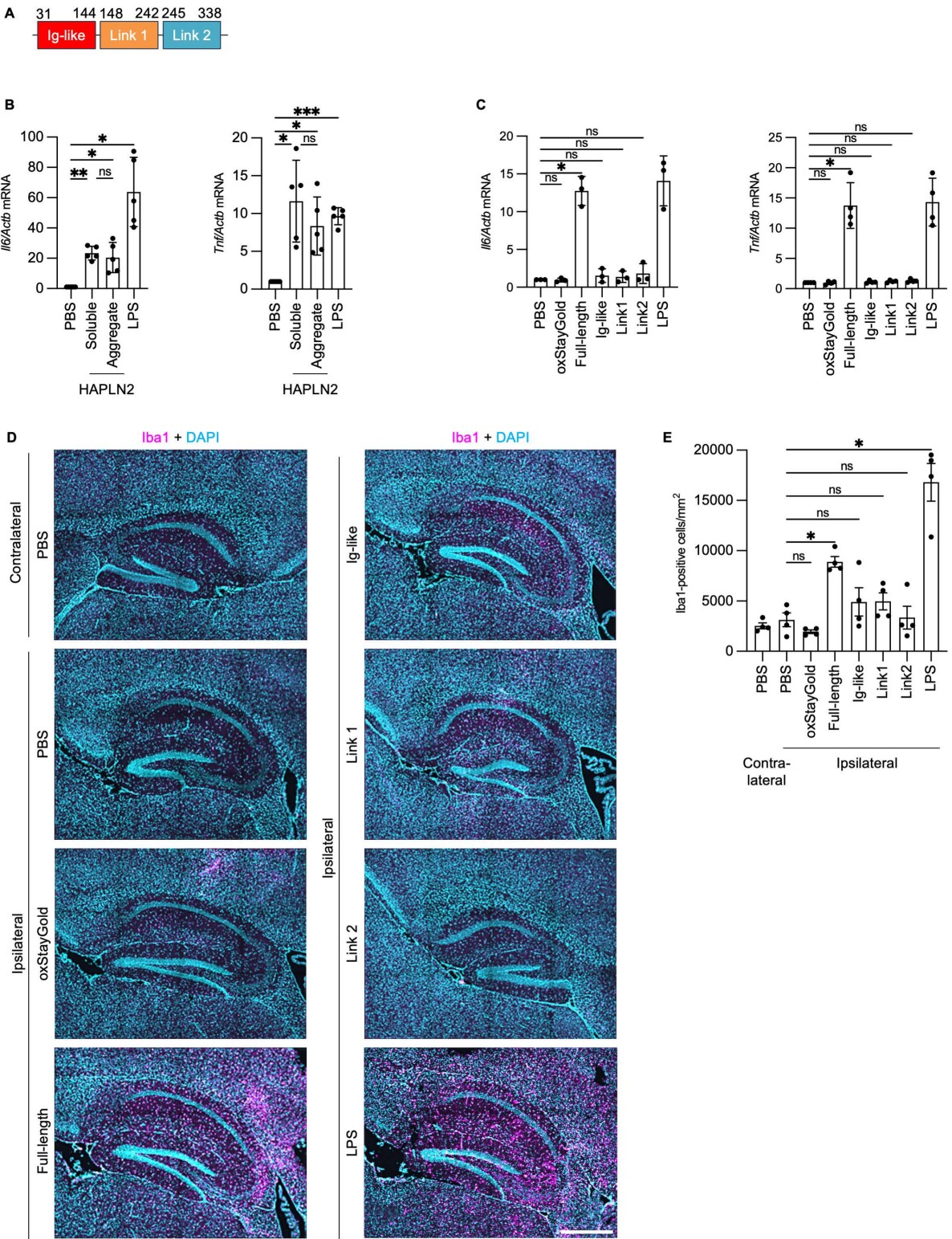

**Fig 8. HAPLN2 induced microglial activation *in vitro* and *in vivo*.** (A) Diagram of the domains within HAPLN2. (B) Reverse transcription polymerase chain reaction (RT-qPCR) analysis of pro-inflammatory cytokine mRNA (*Il6* and *Tnf*, normalized to *Actb*) in MG6 cells stimulated with 1 μM LPS, and 200 nM recombinant full-length HAPLN2 in soluble and aggregated states for four hours. PBS served as a control. Recombinant HAPLN2

was aggregated by incubation in a buffer containing 10 mM sodium phosphate and 300 mM NaCl (pH 6.0) for 24 hours at 37ºC. The oligonucleotide sequences used for quantitative real-time PCR are listed in Table 1. (C) RT-qPCR analysis of pro-inflammatory cytokine mRNA (*Il6* and *Tnf*, normalized to *Actb*) in MG6 cells stimulated with 1 μM LPS, and 200 nM of recombinant proteins (oxStayGold, full-length HAPLN2, HAPLN2 Ig-like domain, HAPLN2 link 1 domain, and HAPLN2 link 2 domain) for four hours. PBS served as a control. (D) Fluorescence immunohistochemistry images of the hippocampus of 3-month-old mice injected with PBS, 20 μg LPS, and 15 pmol of recombinant proteins (oxStayGold, full-length HAPLN2, HAPLN2 Ig-like domain, HAPLN2 link 1 domain, and HAPLN2 link 2 domain). The brains were stained with anti-Iba1 antibody to detect activated microglia. Scale bar: 500 μm. n = 4. (E) Quantitation of the number of activated microglia labeled by Iba1 from (E). Error bars represent mean ± S.D. (C) and S.E.M. (E). P-values were calculated using one-way ANOVA followed by Tukey's *post hoc* tests. *$p < 0.05$, ***$p < 0.001$, ns = not significant. The underlying data for (B), (C), and (E) can be found in S1 Data.

**Table 1. Oligonucleotides used for quantitative real-time PCR.**

| Gene | Forward (5′–3′) | Reverse (5′–3′) |
| --- | --- | --- |
| *Il6* | TGTGAAATCAGGATGCTCTGG | CACTTTATTGGGCTCTATACAATATGC |
| *Tnf* | TCTTCTCATTCCTGCTTGTGG | GGTCTGGGCCATAGAACTGA |
| *Actb* | ACCAGAGGCATACAGGGACA | CTAAGGCCAACCGTGAAAAG |

To determine whether specific domains of HAPLN2 induce pro-inflammatory microglial activation, we added soluble full-length HAPLN2 and its individual domains to MG6 cells. Soluble oxStayGold fluorescent protein, which was not trapped by the nitrocellulose membrane, was used as a negative control (S8A Fig). Soluble full-length HAPLN2 significantly induced IL-6 and TNF-α mRNA expression, but none of the individual domains of HAPLN2 upregulated these cytokines (Fig 8C). These results suggest that the pro-inflammatory effect is associated with full-length HAPLN2 and not with specific domains.

Finally, we examined whether HAPLN2 oligomers induce microglial activation in mouse brains by infusing full-length HAPLN2 and each domain into the hippocampus. Injection of full-length HAPLN2 increased pro-inflammatory microglia, marked by Iba1, in the hippocampus (Figs 8D, 8E, S8B, and S8C). In contrast, oxStayGold and each domain of HAPLN2 did not significantly induce microglial activation, consistent with Fig 8C. These results suggest that oligomeric full-length HAPLN2 promotes microglial activation *in vivo*.

### HAPLN2 accumulated with age in the human cerebellum

Finally, we examined whether HAPLN2 accumulates with age in the human brain. We stained HAPLN2 in the cerebellum of human brains in their sixties or eighties without dementia (S7 Table). We observed the increased HAPLN2 puncta in the human brain in their eighties compared to those in their sixties (Fig 9A). These HAPLN2 puncta did not colocalize with Caspr, suggesting that these are age-dependent aggregates. In addition, the mean count and area of HAPLN2 puncta increased with age (Figs 9B and 9C). These results suggest that HAPLN2 accumulation is a common hallmark of aging in both mice and humans.

## Discussion

Research on age-related protein aggregation in mammals has progressed at a slower pace than that in *C. elegans* due to the difficulty in obtaining a sufficient number of aging individuals. Recently, studies have aimed to identify proteins that become Triton X-100 insoluble in the cerebral cortex with age [75]. However, as demonstrated by the evidence that membraneless organelle constituent proteins and functional large protein complexes, such as the cytoskeleton, also become Triton X-100 insoluble, these proteins are not synonymous with aggregated proteins [75]. Therefore, we used sarkosyl to search for proteins that aggregate with aging because sarkosyl fractionation is known to isolate protein aggregates, including Aβ and tau, into the 100,000 *g* pellet fraction. We identified seven proteins that become sarkosyl-insoluble and accumulate in mouse brains with aging. Of these, five were extracellular matrix components, and a protease, all of which

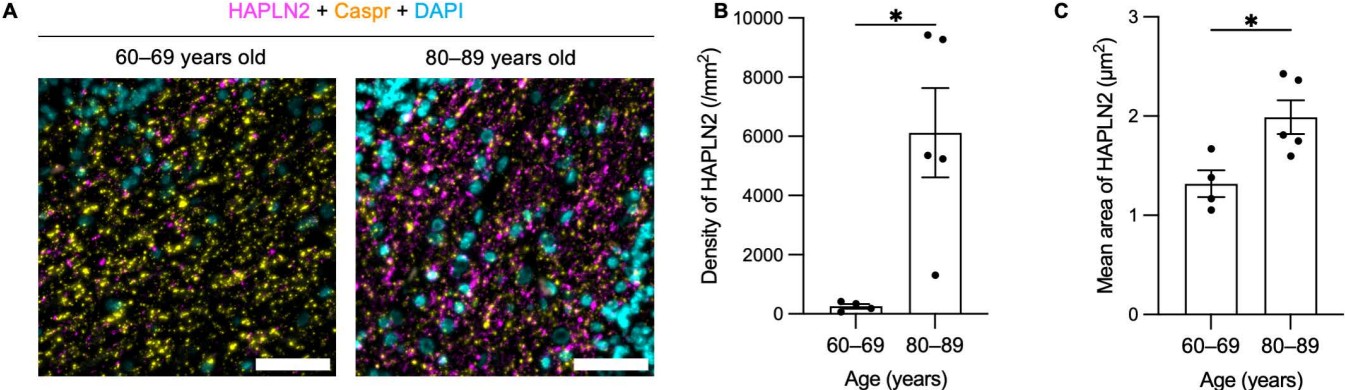

**Fig 9. HAPLN2 formed large puncta with age in the human cerebellum.** (A) Fluorescent immunohistochemistry images for HAPLN2, Caspr, and DAPI in the cerebellum from human post-mortem brains without dementia. Scale bar: 50 μm. (B) Quantification of the mean counts of HAPLN2 puncta in (A). (C) Quantification of the average area of HAPLN2 puncta across all samples in (A). Error bars represent mean ± S.E.M. *P*-values were calculated using two-tailed Student *t* test. *$p < 0.05$. The underlying data for (B) can be found in S1 Data.

are localized extracellularly. We focused on HAPLN2, a component of the perinodal extracellular matrix, as a protein that aggregates in the highly myelinated regions, such as cerebellum white matter and corpus callosum in an age-dependent manner. HAPLN2 aggregates colocalized with hyaluronic acid in the cerebellum white matter in aged mice (Fig 3D). This observation, coupled with the *in vitro* finding that hyaluronic acid promotes aggregation of HAPLN2 (Fig 5A), suggests a key role for hyaluronic acid in this process. This is consistent with reports that the concentration of hyaluronic acid increases, especially in the prefrontal cortex and cerebellum, with age *in vivo* [68,76,77]. Furthermore, it is reported that microglia are fewer in number and more sparsely distributed in the cerebellum compared to other brain regions [78–80]. Therefore, clearance of protein aggregates by microglial phagocytosis may be less efficient in the cerebellum compared to the cerebrum. Moreover, the cerebellum has a significantly higher density of neurons and axons compared to the cerebrum [81]. This implies a greater number of nodes of Ranvier and their associated perinodal extracellular matrix, which could contribute to the high expression of HAPLN2 observed in this region. This may explain the significant HAPLN2 aggregation observed in the cerebellum.

Experiments using recombinant HAPLN2 revealed that the formed amorphous aggregates partially contained cross-β-sheet structures, characteristic of proteinaceous aggregates (Fig 5C). While HAPLN2 binds hyaluronic acid to form a functional perinodal extracellular matrix, these findings indicate that during aggregation, HAPLN2 forms aggregates primarily through protein-protein interactions rather than merely forming hyaluronic acid complexes. Notably, HAPLN2 isolated from the sarkosyl-insoluble P2 fraction of aged mouse brains remained insoluble even after enzymatic digestion of hyaluronic acid (Fig 2D and 2E). This implies that while hyaluronic acid interactions may initiate aggregation, HAPLN2 eventually develops protein-protein interactions that stabilize its sarkosyl-insoluble state.

Adding another layer to the role of hyaluronic acid, we also showed that injecting hyaluronidase into the mouse cerebellum led to a decrease in sarkosyl-insoluble HAPLN2 by promoting its clearance into the CSF (Fig 6). This reduction may be attributed to increased accessibility of clearance systems, such as the glymphatic system, resulting from the expansion of extracellular spaces [69]. This finding is supported by studies showing that increased extracellular volume enhances Aβ clearance [82]. Furthermore, aging reduces CSF flow, and this decline impairs the efficiency of the glymphatic system in clearing extracellular waste [13]. Nonetheless, it remains unclear whether the glymphatic system can effectively transport large protein aggregates, such as HAPLN2 aggregates. Thus, a critical unanswered question is whether HAPLN2 clearance occurs via transport of intact aggregates or requires prior disaggregation into a solubilized form. Future studies will

address this mechanistic ambiguity, which has broad implications for understanding how the glymphatic system handles pathological protein assemblies.

Additionally, recombinant HAPLN2 protein efficiently formed aggregates under low pH conditions (Fig 5). Although the decrease in brain pH with age is relatively small, ranging from 6.30 to 6.00 in humans and from 6.85 to 6.70 in mice [83], it cannot be completely ruled out as a contributing factor to HAPLN2 aggregation. While this modest change in acidity alone may not be sufficient to induce aggregation, it is possible that it interacts with other aging-related factors to promote HAPLN2 aggregation. Taken together, our data suggest that changes in brain pH, hyaluronic acid levels, and the glymphatic flow may contribute to the aggregation of HAPLN2 *in vivo*, highlighting the complex interplay between protein aggregation and the aging brain environment, particularly for HAPLN2.

Microglia play a critical role in clearing extracellular waste, such as protein aggregates, through phagocytosis [71]. Recent studies have also shown that microglia phagocytose the extracellular matrix, including the perineuronal net [84,85]. However, aging is associated with a decline in microglial phagocytic capacity due to increased expression of CD22, a negative regulator of phagocytosis [11,44], and decreased expression of TREM2 and ABCA7, which are well known to mediate phagocytic clearance of Aβ [86,87]. Our study demonstrated that microglial depletion promotes the accumulation of HAPLN2 aggregates (Fig 7), suggesting that age-related impairment of microglial phagocytosis disrupts the clearance of HAPLN2 and the perinodal extracellular matrix. The accumulation of protein oligomers and aggregates in the extracellular space activates microglia. This leads to the secretion of pro-inflammatory cytokines, such as IL-6 and TNF-α, which contribute to chronic inflammation [88]. Our study showed that excessive HAPLN2 oligomers and aggregates activate microglia both *in vitro* and *in vivo* (Fig 8). These findings indicate that while microglia are involved in clearing HAPLN2 aggregates, their phagocytic capacity can become overwhelmed by the abnormal accumulation of HAPLN2 aggregates. This imbalance may trigger microglial pro-inflammatory activation, further contributing to chronic inflammation, a key feature of brain aging. While we demonstrated microglial activation both *in vitro* and *in vivo* following exposure to recombinant HAPLN2 oligomers and aggregates, these findings remain correlative and do not establish a causal relationship, as we did not manipulate endogenous HAPLN2 expression *in vivo*. Therefore, the physiological relevance of HAPLN2 aggregation in the aged brain remains to be fully elucidated. To overcome these limitations, future studies will employ genetic and pharmacological approaches to clarify the causal role of HAPLN2 in neuroinflammation. We believe our current findings provide a solid foundation for these next steps and offer valuable insight into the potential contribution of HAPLN2 to brain aging.

In summary, we have identified HAPLN2, a component of the perinodal extracellular matrix, as an age-dependent protein aggregate. HAPLN2 aggregates mislocalize and accumulate in the extracellular space in aged mouse brains, where hyaluronic acid and microglia regulate their accumulation and clearance. We also observed HAPLN2 aggregates in non-demented human brains. Our study suggests HAPLN2 aggregation as a potential factor in chronic inflammation in the brain during physiological aging and sheds new light on our understanding of age-related decline in brain function.

## Materials and methods

### Post-mortem brain samples

The paraffin-embedded postmortem cerebella derived from neurologically normal patients in their sixties (n = 4) or eighties (n = 5) were obtained from the Japan Brain Bank Net (JBBN). Information on human donors is provided in S7 Table. In accordance with the Declaration of Helsinki, written informed consent was obtained from all participants prior to their passing, or from their families. Experiments using human samples were performed with institutional approval and guidelines from the institutional ethical committee of the JBBN and the Graduate School of Pharmaceutical Sciences, University of Tokyo (No. 4–8).

### Animals

These experiments were conducted in strict accordance with the following Japanese laws and guidelines: the Act on Welfare and Management of Animals (Act No. 105 of October 1, 1973), the Guidelines for Proper Conduct of Animal

Experiments (published by the Science Council of Japan on June 1, 2006), and the Fundamental Guidelines for Proper Conduct of Animal Experiment and Related Activities in Academic Research Institutions (published by the Ministry of Education, Culture, Sports, Science and Technology of Japan, Notice No. 71 of June 1, 2006). C57BL/6N mice were purchased from Clea Japan. All mice were maintained under specific pathogen-free conditions. All animal experiments were performed after obtaining approval from the Institutional Animal Care Committee of the Graduate School of Pharmaceutical Sciences, the University of Tokyo (approval number P26-7). Male mice were used in all experiments. Mice were euthanized by cervical dislocation to preserve the metabolic environment of the brains and to prevent artifacts that could alter the biochemical profiles of the proteome. Mouse brains were bisected down the midline to yield two hemispheres. The brain hemisphere of each animal was quickly frozen on dry ice and stored at −150ºC until use.

### Cell culture

The MG6 cell line is a c-Myc-immortalized cell line of mouse microglia. MG6 cells were maintained at 5% $CO_2$ at 37ºC in Dulbecco's modified Eagle's medium supplemented with 10% (v/v) fetal bovine serum (Thermo Fisher Scientific, 10,270−106), 10 µg/mL insulin (Nacalai Tesque, 12,878−44), and 0.1 mM 2-mercaptoethanol [89,90].

### Label-free quantitative proteomic analysis of sarkosyl-insoluble fraction

The fractionation of the brain tissues was prepared as previously described [31]. Tissues were crushed using Multi-Beads-Shocker (Yasui Kikai, MB601U(S)). Then the tissues were homogenized with a Potter-type homogenizer (AS ONE CORPORATION) at 800 rpm in 1.4 mL of sarkosyl buffer [PBS, 1% sarkosyl (Nacalai Tesque, 20,135–14) and 1 mM phenylmethylsulfonyl fluoride (PMSF) (Wako, 020–15,372)]. The homogenates were centrifuged at 15,000 g for 5 min twice, then for 15 min once at 4°C to obtain supernatant (S1) and pellet (P1) fractions. The supernatants were centrifuged at 100,000 g for one hour at 4°C to obtain sarkosyl-soluble (S2) and sarkosyl-insoluble (P2) fractions. The P2 fraction was resuspended in 140 µL of phase transfer surfactant PTS [12 mM sodium deoxycholate (SDC), 12 mM sarkosyl, and 200 mM ammonium bicarbonate] buffer. The resuspended P2 solutions were sonicated with Bioruptor Plus (BMBio, B01020001) for 10 cycles, (30"/30" cycles), and incubated for 30 min at 60°C with 5 mM dithiothreitol (DTT) for solubilization and reduction of the proteins. The samples were then alkylated with 200 mM methyl methanethiosulfonate for 30 min at room temperature in the dark. The samples were purified with methanol-chloroform extraction and solubilized with 50 µL of Rapigest buffer [0.1% Rapigest SF (Waters, 186,002,123) and 50 mM tetraethylammonium bicarbonate (Thermo Fisher Scientific, 90,114)]. The alkylated samples were sonicated and were digested overnight with 1 µg of Trypsin Gold (Promega, V5280) at 37°C. Peptides were purified from the supernatant by GL-tip SDB (GL Sciences, 7,820–11,200) and eluted with QE-B solution [70% Acetonitrile and 0.1% formic acid] (Kanto Chemical Co. , 01922–64). Peptides were concentrated with Concentrator Plus for 45°C and resuspended with 30 µL of QE-A solution [0.1% formic acid] (Kanto Chemical Co. , 16,245–12). The concentration of peptides was measured with Pierce Quantitative Colorimetric Peptide Assay (Thermo Fisher Scientific, 23,275). 500 ng of peptides from each fraction were subjected to proteomic analysis using an EASY-nLC 1,200 system connected online to Q-Exactive (Thermo Fisher Scientific). Samples were loaded onto Acclaim PepMap 100 C18 column (75 µm × 2 cm, 3 µm particle size and 100 Å pore size; 164,946, Thermo Fisher Scientific) and separated using capillary column, (75 µm × 12.5 cm, 3 µm particle size, NTCC-360/75-3-125, Nikkyo Technos) in an EASY-nLC 1,200 system (Thermo Fisher Scientific). Raw data were analyzed using Proteome Discoverer (version 2.4, Thermo Scientific). Quantification was performed at the precursor ion level, and data were normalized using the "Total Peptide Amount" normalization method. The mass tolerances for the precursor and fragment ions were 10 ppm and 0.02 Da, respectively, and peptide identification was filtered at a false discovery rate < 0.01. Data were processed using the R package MSnbase [91]. The missing values were computed using the method of minDet by replacing values with the minimal value observed in the sample. The mass spectrometry proteomics data have been deposited to the ProteomeXchange Consortium via the jPOST partner repository (https://repository.jpostdb.org/) with the dataset identifier JPST003565 and PXD060043 [92,93].

## Label-free proteomics of the PTS-soluble fraction

Tissues were crushed and homogenized in 400 µL of PTS buffer. The homogenates were centrifuged at 15,000 $g$ for 15 min at 4°C to yield the supernatant and pellet fractions. Thirty µg of the supernatant was reduced with 5 mM DTT for 15 min at room temperature. The samples were then alkylated and digested by Trypsin Gold. The samples were centrifuged at 15,000 $g,$ and the supernatants were added with 0.1% trifluoroacetate (Wako, 192−09912), incubated for 45 min at 37°C, and centrifuged at 15,000 $g$ for 15 min at 4°C. Peptides were purified from the supernatant by GL-tip SDB (GL Sciences, 7,820−11,200) and eluted with QE-B solution [70% Acetonitrile and 0.1% formic acid] (Kanto Chemical Co., 01922−64). The eluates were concentrated at 45°C and resuspended with 40 µL of QE-A. 500 ng of peptides were centrifuged at 15,000 $g$ for 15 min at 4°C and subjected to proteomic analysis with Q-Exactive. The mass spectrometry proteomics data have been deposited to the ProteomeXchange Consortium via the jPOST partner repository (https://repository.jpostdb.org/) with the dataset identifier JPST003863 and PXD064908 [92,93].

## GO-term analysis

In the GO analysis of proteomic data, we used the Database for Annotation, Visualization and Integrated Discovery (DAVID) [94].

## Antibodies

Primary antibodies for immunoblotting (IB) and immunohistochemistry (IHC) are listed in S8 Table. All secondary antibodies used for immunoblotting were purchased from Jackson ImmunoResearch Laboratories: Peroxidase AffiniPure Rabbit Anti-Mouse IgG + IgM (H + L) (315-035-048), Peroxidase AffiniPure Goat Anti-Rabbit IgG (H + L) (111-035-144), and Peroxidase AffiniPure Donkey Anti-Goat IgG (H + L) (705-035-147). All the secondary antibodies used for immunohistochemistry were purchased from ThermoFisher Scientific: Alexa Fluor 488 Goat Anti-Mouse IgG (H + L) (A-28175), Alexa Fluor 488 Goat Anti-Rabbit IgG (H + L) (A-11008), Alexa Fluor 555 Mouse Anti-Mouse IgG (H + L) (A-28180), Alexa Fluor 647 Streptavidin (S21374).

## Immunoblot analysis

Sample preparation, sodium dodecyl sulfate-polyacrylamide gel electrophoresis (SDS-PAGE), and immunoblot analysis were performed as described previously [95]. Each fraction of the brain lysates was mixed with SDS lysis buffer [50 mM Tris-HCl (pH 8.0), 150 mM NaCl, 0.1% SDS, 1% Triton X-100, 0.5% SDC, and 1 mM PMSF] and centrifuged at 20,000 $g$ for 20 min at 4°C. The supernatants were separated by SDS-PAGE (12.5% acrylamide, 7 × 8 cm, 1 mm thick). Proteins were electrotransferred to Immobilon-P polyvinylidene difluoride (PVDF) (Merck Millipore, IPVH00010), at 30 V for one hour at room temperature. For immunoblot analysis, the membranes were incubated with Blocking One (Nacalai Tesque, 03953–95) for one hour at room temperature. The membranes were incubated with primary antibodies in TBS-T buffer [20 mM Tris-HCl (pH 8.0), 150 mM NaCl, and 0.1% Tween-20] overnight at 4°C, washed with TBS-T buffer and incubated with horseradish-peroxidase conjugated secondary antibody (1:20,000) for 30 min at room temperature. The membranes were immersed with ImmunoLightning Plus (PerkinElmer, NEL105001EA) or ImmunoSTAR LD (Wako, 290–69,904), and the images were acquired with FUSION FX7.EDGE (VILBER). For band quantification, the intensity of each band was measured using Fiji software (ImageJ2, version 2.14.0/1.54h) [96].

## Dual membrane filter trap assay

The S1 fractions from young and aged mouse brains were subjected to 2-fold serial dilution using sarkosyl buffer. Fifty microliters of each diluted sample were loaded onto a pre-wet 0.45-µm pore size nitrocellulose membrane (Bio-Rad, 1620167), which was layered over a pre-wet Immobilon-P (PVDF) membrane (0.2-µm pore size) in a Bio-dot

Microfiltration Apparatus (Bio-Rad, 1706545). Vacuum filtration was performed under controlled conditions to ensure sequential passage of the protein solution through the nitrocellulose membrane and then through the PVDF membrane. Each well underwent three washing cycles with sarkosyl buffer. Subsequently, both membranes were processed using standard immunoblotting protocols starting from the blocking step onward.

## Immunohistochemical analysis

The paraffin sections (10 µm) were boiled in 10 mM sodium citrate buffer (pH 6.0) for one minute in a microwave oven for heat-induced antigen retrieval. The sections were incubated for one hour in Blocking One Histo (Nacalai Tesque, 06349−64), then overnight at 4°C with primary antibodies diluted in Blocking One Histo. After rinsing, sections were incubated with secondary antibodies and 1 µg/mL 4′,6-diamido-2-phenylindole dihydrochloride (DAPI) (Nacalai Tesque, 11,034−56) diluted in Blocking One Histo for one hour at room temperature. The sections were washed by TBS-T three times and by PBS three times, then immersed in TrueBlack Lipofuscin Autofluorescence Quencher (Biotium , 23,007) for 20 min to reduce autofluorescence from lipofuscin. The slices were mounted with Prolong Glass Antifade Mountant (Invitrogen, P36980).

For the staining of Iba1, TSA plus amplification kit (Akoya Biosciences, NEL741001KT) was used in combination with subsequent conventional immunofluorescence staining for other antigens. To block endogenous peroxidase activity, brain sections were treated with 3% $H_2O_2$ in methanol for 15 min at room temperature. After washing, sections were treated with Blocking one Histo for one hour at room temperature, and then incubated with 0.1% anti-Iba1 antibody (Wako, 013–27,691) in 5% Blocking one Histo in TBS-T overnight at 4°C. The sections were incubated with 0.01% Peroxidase AffiniPure Goat Anti-Rabbit IgG (H + L) (Jackson ImmunoResearch Laboratories, 111-035-144) diluted in TNB buffer at 4°C overnight. After washing with TNT buffer (0.1% Triton X-100, 100 mM Tris-HCl (pH 7.5), 150 mM NaCl) three times, the sections were reacted with Fluorescein Plus amplification reagent for 10 minutes at room temperature, washed with TNT.

For ACAN staining, brain sections were treated with chondroitinase ABC (ChABC, 0.1 U/ml; Sigma, C2905) at 37°C for three hours before blocking. Images were captured by BZ-X800 (Keyence) using 100 X objectives or Leica DMi8 Thunder (Leica Microsystems) using 10X or 63X objectives. The images were processed and analyzed with Fiji software. Colocalization analyses were performed with the JaCoP plugin in Fiji software [97].

## Protein purification

Recombinant HAPLN2 was purified using the method to purify the versican core protein G1 domain as previously described [98]. The HAPLN2 coding sequence, nucleotides 82–1,026 was amplified from mouse brain cDNA by PCR and cloned into pGEX6p-1 (27-4597-01; Addgene). The glutathione-S-transferase tag was removed by PCR. The construct was transformed into ClearColi BL21(DE3) Electrocompetent Cells (60,810−1, LGC Biosearch Technologies). The cells were grown in LB Miller broth containing 100 µg/mL ampicillin at 37°C with shaking at 150 rpm. The *E. coli* cultures were induced with 0.8 mM isopropyl β-D-thiogalactopyranoside (06289−67; Nacalai Tesque), grown for another 20 hours at 20°C, and harvested by centrifugation at 10,000 *g* for 20 min at 4°C (Avanti J-25; Beckman Coulter). Inclusion bodies were purified and solubilized in urea and the protein was refolded, essentially as described previously [98]. Briefly, inclusion bodies were isolated with an *E. coli* lysis buffer [1% (w/v) Nonidet P-40, 1% (w/v) Triton X-100, 1 mM PMSF, and 1 mM DTT in PBS] solubilized in a denaturation buffer [8 M urea in 50 mM Tris-HCl (pH 7.5)] and clarified by centrifugation. The supernatant was purified by incubating with Ni-NTA agarose (QIAGEN, 30,210) overnight and eluted with elution buffer [6 M urea, 500 mM imidazole, and 10 mM HCl] and then diluted 66-fold into refolding buffer [1 mM ethylenediaminetetraacetic acid, 500 mM L-arginine, 1 mM cysteine, 2 mM cystine, and 20 mM ethanolamine (pH 11.0)] by dropwise addition at 4°C with gentle stirring, followed by incubation overnight without stirring. After refolding, the protein was concentrated to 2 mL with Amicon Ultra-15 Centrifugal Filter 10 kDa MWCO (Millipore, 36,100,101), and dialyzed overnight into a storage buffer [10 mM sodium phosphate (pH 8.0)].

## Protein aggregation assay

Recombinant HAPLN2 proteins (500 nM) were subjected to a series of pH (6.0–8.0), NaCl concentration (0–300 mM), and hyaluronic acid (0–1 mg/mL) in a total volume of 100 μL. The samples were deposited on μ-Slide 18-Well Glass Bottom (Ibidi, 81,817) and incubated for 24 hours at 37ºC before being imaged on Eclipse Ti2 microscope (Nikon Corporation) with Micromanager software [99] in which bright field images were taken using a 20X air magnification objective. Aggregate formation was imaged by collecting a series of images in the bright field channel. Fiji software was used in all image processing.

## *In vitro* single-aggregate imaging

Super-resolution imaging was conducted based on point accumulation for imaging nanoscale topography (PAINT). Borosilicate glass coverslips (VWR International, 22 × 22 mm, 631−0124) were cleaned with an argon plasma cleaner (Harrick Plasma, PDC-002) for at least one hour to remove fluorescent residues. Before use, each batch of cover slides was tested for fluorescent artifacts. A drop of 10 μL of the mixture of HAPLN2 used in the protein aggregation assay was placed over a coverslip together with Thioflavin T and Nile Red dyes diluted in PBS to achieve final concentrations of 5 μM for Thioflavin T and 5 nM for Nile Red. Imaging was performed using a CFI Apochromat TIRF 100x Oil immersion objective (Nikon Corporation), mounted on an Eclipse Ti2 microscope (Nikon Corporation). Thioflavin T was excited using a 405 nm laser (Cobolt #0405-06-01-0120-100) with a beam power of 20 mW. Emitted fluorescence was collected by the same objective and separated from the excitation light by a dichroic (Semrock #DI03-R405/488/532/635-T1-25X36), subsequently passed through an emission filter (Semrock #FF01–510/84–25) and focused onto a CMOS camera (ORCA-Flash4.0 V3 Digital CMOS; Hamamatsu). Nile Red was excited by a 532 nm laser (Cobolt #0532-06-91-0100-100) with a beam power of 80 mW. Both lasers were positioned at a highly inclined and laminated optical sheet (HiLo) illumination angle to image large amorphous aggregates at the surface of the coverslip (exciting ~5 μm deep in the sample) while minimizing background fluorescence. Emitted light was separated using a dichroic (DC/ZT532rdc-UF2; Chroma) and passed through an emission filter (Semrock #FF01–650/150–25) before being focused onto the CMOS camera. The image pixel size was 65 nm. A z-stack of 5.0 μm (10 planes with 0.5 μm spacing) was acquired in all channels. The microscope was controlled with Micromanager software, and the frame interval was 50 ms. Each analyzed image corresponds to an average of 10 images in the bright-field channel, 50 images in the thioflavin T, and 1,000 images in the Nile Red channel. The Nile Red images were reconstructed with the ThunderSTORM plug-in in Fiji software [100]. The morphological analysis of Nile Red-positive structures from reconstructed images was conducted by LocAlization Microscopy Analyzer (LAMA) [59] and MATLAB (Mathworks, ver. R2023b).

## Circular dichroism (CD) spectroscopy

CD spectra were recorded using a J-820 CD spectrometer (JASCO) at room temperature. Recombinant HAPLN2 protein (400 ng/μl) was used in its soluble state, and aggregation was induced by adjusting the pH to 6.0 with HCl, followed by overnight incubation at 37°C. The aggregated samples were sonicated for 10 minutes prior to measurement. To minimize buffer absorption, the 400 ng/μl samples were diluted 1:8 with deionized water. CD spectra were acquired from 190 to 250 nm at a scan rate of 50 nm/min, with a data pitch of 0.1 nm. For each sample, three to four spectra were averaged and smoothed using binomial approximation. Secondary structure content was estimated using BeStSel [61].

## Negative-stained transmission electron microscopy

A volume of 1.5 μL of recombinant samples was deposited onto glow-discharged formvar-coated 300-mesh copper grids (Stork Veco) for 20 min at room temperature. The grids were blotted with filter paper (Toyo Roshi Kaisha, , Tokyo, Japan), washed twice with PBS, washed twice with distilled water, stained with 2% (w/v) uranyl acetate for 30 seconds, blotted and air-dried. The prepared specimens were finally examined with an electron microscope (HT7700; Hitachi) at 100 kV.

## Stereotaxic injection

Mice were anesthetized by intraperitoneal injection with 5.0 mL/kg of a non-narcotic anesthetic combination (Me/Mi/Bu) [30 µg/mL medetomidine (Zenoaq), 0.4 mg/mL midazolam (Astellas), and 0.5 mg/mL butorphanol (Meiji Seika Pharma)] and fixed in a stereotaxic surgery frame.

For intracerebellar injection, 20 mg/mL hyaluronidase from ovine testes (FUJIFILM Wako Pure Chemical Corporation, 37326-33-3) in a total volume of 1.0 µL was injected at the following coordinates relative to the bregma: anterior-posterior (A/P) −5.5 mm medial-lateral (M/L) −1.2 mm, and dorsal-ventral (D/V) −2.5 mm. Mice were sacrificed one day post-surgery.

For intrahippocampal injection, either 15 pmol of protein or 5 mg/mL LPS solution, each in a total volume of 2.0 µL, was injected at the following coordinates relative to the bregma: A/P −1.7 mm, M/L −1.6 mm, and D/V −1.9 mm. Mice were sacrificed one week post-surgery.

## CSF collection

CSF was collected as described in previous studies [101]. Briefly, mice were anesthetized with a non-narcotic anesthetic combination (Me/Mi/Bu), and the head was secured using a stereotaxic frame (Narishige, SGM-4). The mouse's head was then lowered to create a 45° angle between the cisterna magna and the nose. Under a dissection microscope, the scalps and the three overlapping layers of neck muscle were carefully dissected to expose the cisterna magna. A borosilicate glass capillary with filament (Narishige, GD-1.2) was prepared by pulling with a micropipette puller (Narishige, PC100) and connected to a 1 mL syringe via polyethylene tubing (Saint-Gobain, LMT-55). The capillary was mounted onto a three-dimensional micromanipulator (Narishige, MM3) and gently inserted into the cisterna magna. Using the micromanipulator, the capillary was inserted into the cisterna magna and then retracted until CSF began to flow into the capillary. Negative pressure was applied until approximately 10 µL of CSF was collected, after which the capillary was fully withdrawn from the cisterna magna. Throughout the experiment, the body temperature of the mouse was maintained at 37°C using a temperature control system (Muromachi Kikai Co., , NS-TC10).

## Capillary western blot of HAPLN2 in CSF

Capillary western blot analysis was conducted using Jess (Protein Simple). A volume of 1 µL of cerebrospinal fluid was used for the experiment. Band intensities were calculated using Fiji software.

## Label-free proteomics of the CSF

From 5 µL of CSF, proteins were extracted by methanol-chloroform precipitation and the resulting pellets were resuspended in 50 µL of PTS-buffer. The samples were reduced with 5 mM DTT for 15 min at room temperature. The samples were then alkylated and digested by Trypsin Gold. The samples were centrifuged at 15,000 $g$, and the supernatants were added with 0.1% trifluoroacetate (Wako, 192−09912), incubated for 45 min at 37°C, and centrifuged at 15,000 $g$ for 15 min at 4°C. Peptides were purified from the supernatant by GL-tip SDB (GL Sciences, 7,820−11,200) and eluted with QE-B solution [70% Acetonitrile and 0.1% formic acid] (Kanto Chemical Co. , 01922−64). The eluates were concentrated at 45°C and resuspended in 40 µL of QE-A. 500 ng of peptides was centrifuged at 15,000 $g$ for 15 min at 4°C and subjected to proteomic analysis with Q-Exactive.

## Open field test

The behavior of mice was recorded using the Noldus system and analyzed with EthoVision XT software (Noldus Information Technology). The test was conducted in a high-walled open-field box (30 × 30 cm), and the animals were observed for five minutes.

## Depletion of microglia by PLX3397 treatment

PLX3397 [72] was synthesized as described previously [102] and formulated at 290 mg/kg in AIN-76A standard chow by Research Diets The chows were given to 4-week-old wild-type mice (C57/BL6J) for two months. Depletion of microglia was confirmed by immunohistochemistry for the microglial marker Iba1.

## RNA isolation and quantitative real-time PCR

Total RNA was extracted from MG6 cells grown in 12-well tissue culture plates ($7 \times 10^4$ cells/well) by High Pure RNA Isolation Kit (Roche, 11828665001). The isolated total RNA was then reverse-transcribed with ReverTra Ace (TOYOBO, FSQ-201). cDNAs were amplified with THUNDERBIRD Probe qPCR mix (TOYOBO, QPS-101) and Universal Probe Library probe (Roche) with the following specific primer sets. mRNA levels were measured with LightCycler 480 (Roche).

## Statistics and reproducibility

All data were derived from at least three independent experiments and are presented as mean ± S.D. or ± S.E.M. as stated in the figure legends. Statistical analyses were performed with GraphPad Prism 9.0 software. An unpaired two-tailed Student t test or one-way ANOVA followed by Tukey's *post hoc* test was performed as described in the figure legends. All replication attempts were successful with similar results.

## Supporting information

**S1 Fig. Protein extraction by phase transfer surfactant (PTS) buffer.** Coomassie brilliant blue staining of young (3-month-old) and aged (28-month-old) mouse brains. Whole brains were lysed with PTS buffer and centrifuged at 15,000 *g*. n = 3.
(TIFF)

**S2 Fig. Detailed immunohistochemistry analysis of HAPLN2 in the mouse brain.** (A) Immunohistochemistry images of the mouse whole brain, cerebellum, and corpus callosum stained with anti-HAPLN2 antibody and anti-myelin basic protein (MBP) antibody. n = 3. (B) Magnified immunohistochemistry images of the mouse cerebellar white matter stained with anti-HAPLN2 antibody or biotinylated hyaluronic acid-binding protein (HABP), which probes hyaluronic acid. Specific staining of HAPLN2 and HABP was observed without crossover. n = 3. (C, D) Fluorescence immunohistochemistry images of the mouse cerebellar white matter showing staining for ubiquitin (C) and p62 (D). n = 3. Scale bars: 1 mm (A), 50 µm (B, C, D), and 10 µm (A (inlet)).
(TIFF)

**S3 Fig. The association between age-dependent demyelination and HAPLN2 aggregation in the mouse brain.** (A) Immunohistochemistry images of the mouse cerebellum stained with anti-HAPLN2 antibody, anti-calbindin 1 antibody, and anti-MBP antibody. n = 3. (B) Magnified fluorescence immunohistochemistry images for HAPLN2, calbindin 1, and MBP corresponding to the boxed areas in (A). Regions indicated by magenta arrows showed MBP staining voids accompanied by increased calbindin 1 staining. (C) Quantitation of the mean area of calbindin 1 in (B) comparing young (3-month-old) and aged (18-month-old) mice. (D) Magnified fluorescent immunohistochemistry images of calbindin 1–high and –low regions in the cerebellar white matter of 18-month-old mice. High and low regions were defined as those with more than 5% and less than 1% calbindin 1–positive area, respectively. (E) Quantitation of the mean area of calbindin 1 (left) and HAPLN2-positive puncta (right) in calbindin 1–high region and –low region in 18-month-old mice, as shown in (D). Scale bars: 1 mm (A), 50 µm (B), and 10 µm (B (inlet), D). Error bars represent mean ± S.D. *P*-values were calculated using two-tailed Student t test. **$p < 0.01$, ****$p < 0.0001$, ns = not significant. The underlying data for (C) and (E) can be found in S1 Data.
(TIFF)

**S4 Fig. Age-dependent aggregation of HAPLN1 in the mouse cerebellar nuclei.** (A) Immunohistochemistry images of the mouse cerebellum stained with anti-HAPLN1 antibody and anti-ACAN antibody. n = 3. (B) Magnified fluorescence immunohistochemistry images for HAPLN1 and ACAN corresponding to the boxed areas in (A). The magenta arrows indicate HAPLN1 that are mislocalized relative to ACAN staining. (C) Quantitation of the mean area of HAPLN1 (left) and ACAN-positive area (right) in (B). The underlying data can be found in S1 Data. Scale bars: 1 mm (A), 50 μm (B). Error bars represent mean ± S.D. *P*-values were calculated using two-tailed Student *t* test. ***$p < 0.001$, ns = not significant. (TIFF)

**S5 Fig. Aggregation propensity of recombinant HAPLN2 protein.** (A, B) Brightfield microscopic images showing aggregates of recombinant HAPLN2 incubated under various buffer conditions in the absence (A) or presence (B) of hyaluronic acid. Data were collected from at least five fields of view. (C) Histogram depicting the area of individual aggregates of recombinant HAPLN2 after 24 hours of incubation in the indicated buffer. Aggregate areas were measured using ImageJ (version 1.53t). The underlying data can be found in S1 Data. (D) Transmission electron microscopy images of recombinant HAPLN2 aggregates formed under 150 mM NaCl (pH 6.0). Data were collected from at least five fields of view. (E) Schematic representation of cross-β-sheet-positive oligomers and fibrils stained with thioflavin T and Nile Red. Created with BioRender.com. Scale bars: 10 μm (A, B), 1 μm (D, black), and 100 nm (D, white). (TIFF)

**S6 Fig. Clearance of cerebellar HAPLN1 after intracerebellar hyaluronidase injection.** (A) Immunoblot analysis of the cerebellum corresponding to the samples shown in Fig 6D. n = 4. (B) Densitometric quantification of (A). HAPLN1 protein levels in the S2 fraction were normalized to α-tubulin in the S2 fractions. HAPLN1 protein levels in the P2 fraction were normalized to α-tubulin in the S1 fractions. n = 4. The underlying data can be found in S1 Data. Error bars represent mean ± S.E.M. *P*-values by one-way ANOVA followed by Tukey's *post hoc* test. ***$p < 0.001$, ****$p < 0.0001$, ns = not significant. (TIFF)

**S7 Fig. Association between HAPLN2 accumulation induced by PLX3397 treatment and demyelination. The mean area of calbindin 1 did not show a significant difference after PLX3397 treatment.** (A) Immunohistochemistry images of the mouse cerebellum stained with anti-Iba1 antibody, anti-calbindin 1 antibody, and anti-MBP antibody. n = 3. (B) Magnified fluorescence immunohistochemistry images for Iba1, calbindin 1, and MBP corresponding to the boxed areas in (A). (C) Quantitation of the mean area of calbindin 1 (B). The underlying data can be found in S1 Data. Scale bars: 1 mm (A), 50 μm (B), and 10 μm (B, inlet). Error bars represent mean ± S.D. *P*-values were calculated using two-tailed Student *t* test. ns = not significant. (TIFF)

**S8 Fig. Microglial activation induced by injecting recombinant HAPLN2 protein into the mouse hippocampus.** (A) Filter trap assay and immunoblot analysis of recombinant full-length HAPLN2, HAPLN2 Ig-like domain, and oxStayGold proteins. These recombinant proteins (2 μM each) were incubated overnight at 37°C, followed by centrifugation at 100,000 *g* for one hour at 4°C. (B) Immunohistochemistry images of the coronal brain section from Fig 8E, stained with anti-Iba1 antibody to probe activated microglia. (C) Magnified fluorescence immunohistochemistry images for Iba1, corresponding to Fig 8D. Scale bars: 500 μm (B) and 50 μm (C). (TIFF)

**S1 Table. Label-free proteomics data of the sarkosyl-insoluble fraction from young and aged mouse brains related to Fig 1A.** (XLSX)

**S2 Table. Imputed data of label-free proteomics of the sarkosyl-insoluble fraction of young and aged mouse brains corresponding to S1 Table.** (XLSX)

**S3 Table. 57 proteins enriched in the sarkosyl-insoluble P2 fraction of aged mouse brains in Fig 1A.**
(XLSX)

**S4 Table. Label-free proteomics data of the PTS-soluble fraction from young and aged mouse brains related to Fig 1C.**
(XLSX)

**S5 Table. Imputed data of label-free proteomics of the sarkosyl-insoluble fraction of young and aged mouse brains corresponding to S2 Table.**
(XLSX)

**S6 Table. 27 proteins enriched in the PTS-soluble supernatant of aged mouse brains in Fig 1C.**
(XLSX)

**S7 Table. Human donors applied for the immunohistochemical analysis in Fig 9.** The stages of neurodegenerative diseases in post-mortem brain samples were examined using the following criteria. AT8: Tangle pathology (AT8 Stage) [103], CERAD: The Consortium to Establish a Registry for Alzheimer's Disease, assessing neuritic plaques [104], Thal: Amyloid-beta deposition (Thal Amyloid Phase) [105], DLB (4th Consensus): The Dementia with Lewy Bodies criteria (4th Consensus Report) [106], LB Braak: Lewy body pathology stages (Braak Staging for Lewy Bodies) [107], Grain: Argyrophilic grain pathology (Saito Scale) [108], Joseph-TDP: TDP-43 proteinopathy (Josephs' Staging) [109].
(XLSX)

**S8 Table. Primary antibodies used in immunoblot and immunohistochemistry analyses.**
(XLSX)

**S1 Data. Raw data and corresponding summary statistics supporting each figure.**
(XLSX)

**S1 File. Raw images.** Raw images for all IB.
(PDF)

## Acknowledgments

We thank members of the Laboratory of Morphology and Image Analysis, Biomedical Research Core Facilities, and Juntendo University Graduate School of Medicine for technical assistance with microscopy. ChABC was generously provided by Dr. Shinji Miyata (Tokyo University of Agriculture and Technology). We are grateful to Mr. Hayato Etani for instruction in stereotaxic injection. Figs 6A and S5E were created with BioRender.com (https://www.biorender.com/).

## Author contributions

**Conceptualization:** Shoshiro Hirayama, Shigeo Murata.

**Data curation:** Ayaka Watanabe, Shoshiro Hirayama, Itsuki Kominato, Taeko Kimura, Terunori Sano, Masaki Takao.

**Formal analysis:** Ayaka Watanabe, Sybille Marchese.

**Funding acquisition:** Shoshiro Hirayama, Masaki Takao, Masato Koike, Juan Alberto Varela, Taisuke Tomita, Shigeo Murata.

**Investigation:** Ayaka Watanabe, Shoshiro Hirayama, Itsuki Kominato, Hiroshi Kameda.

**Methodology:** Juan Alberto Varela, Taisuke Tomita.

**Project administration:** Shoshiro Hirayama, Shigeo Murata.

**Resources:** Sybille Marchese, Pietro Esposito, Vanya Metodieva, Terunori Sano, Masaki Takao, Sho Takatori, Masato Koike.

**Software:** Sybille Marchese.

**Supervision:** Shoshiro Hirayama, Juan Alberto Varela, Taisuke Tomita, Shigeo Murata.

**Writing – original draft:** Ayaka Watanabe, Shoshiro Hirayama, Taeko Kimura, Hiroshi Kameda.

**Writing – review & editing:** Itsuki Kominato, Sybille Marchese, Pietro Esposito, Vanya Metodieva, Sho Takatori, Juan Alberto Varela, Shigeo Murata.

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
