## [Editor Report · Decision Letter 0]

7 Jan 2025

Dear Dr Murata,

Thank you for submitting your manuscript entitled "Aggregation of HAPLN2, a component of the perinodal extracellular matrix, is a hallmark of physiological brain aging in mice" for consideration as a Research Article by PLOS Biology. I would like to first apologize for the time it has taken to send you an initial decision - the PLOS Biology editorial office was closed over the holidays which caused the delay.

Your manuscript has now been evaluated by the PLOS Biology editorial staff as well as by an academic editor with relevant expertise and I am writing to let you know that we would like to send your submission out for external peer review.

Once your full submission is complete, your paper will undergo a series of checks in preparation for peer review. After your manuscript has passed the checks it will be sent out for review. To provide the metadata for your submission, please Login to Editorial Manager (https://www.editorialmanager.com/pbiology) within two working days, i.e. by Jan 09 2025 11:59PM.

Kind regards,

Luke

Lucas Smith, Ph.D.

Senior Editor

PLOS Biology

lsmith@plos.org

---

## [Decision Letter · Decision Letter 1]

19 Feb 2025

Dear Dr Murata,

Thank you for your patience while your manuscript "Aggregation of HAPLN2, a component of the perinodal extracellular matrix, is a hallmark of physiological brain aging in mice" was peer-reviewed at PLOS Biology. It has now been evaluated by the PLOS Biology editors, an Academic Editor with relevant expertise, and by several independent reviewers.

In light of the reviews, which you will find at the end of this email, we would like to invite you to revise the work to thoroughly address the reviewers' reports. As you will see below, the reviewers agree that the study is potentially interesting and generally well done. However they have each raised a number of important points aimed at strengthening the study further and we think these will need to be thoroughly addressed before we can consider your paper for publication.

Given the extent of revision needed, we cannot make a decision about publication until we have seen the revised manuscript and your response to the reviewers' comments. Your revised manuscript is likely to be sent for further evaluation by all or a subset of the reviewers.

**IMPORTANT - SUBMITTING YOUR REVISION**

*Re-submission Checklist*

*Published Peer Review*

*PLOS Data Policy*

*Blot and Gel Data Policy*

Sincerely,

Luke

Lucas Smith, Ph.D.

Senior Editor

PLOS Biology

lsmith@plos.org

REVIEWS:

Reviewer #1: In this paper Watanabe et al. investigate what types of detergent insoluble protein aggregates form with increasing age. Using sarkosyl to restrict themselves to insoluble proteins, and then mass spectrometry, they find an abundance of proteoglycans, especially HAPLN2. They further investigate and document the formation of these aggregates biochemically. Using histology they show that with increasing age there appear HAPLN2 aggregates in the brain, consistent with reduced solubility. They report the source of HAPLN2 is likely oligodendrocytes, although they don't perform any experiments to confirm this. Interestingly, the appearance of these aggregates appears not to be correlated with the normal distribution and expression pattern of HAPLN2 at nodes of Ranvier. Importantly, they show that treatment with hyaluronidase reduces the aggregation, and depletion of microglia increases the number and size of the HAPLN2 aggregates. Their work suggests that these aggregates induce inflammation by microglia and these microglia clear these aggregates. Finally, the authors show these same aggregates also exist in the aged human cerebellum.

The work as presented is an excellent descriptive analysis of how HAPLN2 accumulates with aging, and how it activates microglia to induce inflammation and clear these aggregates. This is potentially quite interesting and important. The quality of the data are very high and the results are compelling. I support publication. With that said, this study could be improved and made much more significant by showing that these aggregates have any physiological function. For example, do HAPLN2 knockouts have improved outcomes with increasing age? Although the authors focus on HAPLN2, perhaps there are many other proteins that are equally pathogenic. It would be interesting to know what else is in those aggregates - the authors may already know based on their mass spectrometry results. Although the experiments showing that microglia are central to the process, there are no experiments that show the accumulation of HAPLN2 is biologically significant. I strongly suspect it is, but there aren't any experiments to confirm this.

Minor comments:

1. Line 240: granule cell layer, not granular layer

2. Line 275: Caspr is a component of the paranodal junction, not node. HAPLN2 is localized to the node between the flanking paranodal junctions labeled by Caspr.

Reviewer #2: In this manuscript, Watanabe A et al. assessed which proteins become more insoluble upon aging in the male mouse brain. A decline in proteostasis has long been thought to play a major role in the decline in health upon aging, but few studies have been conducted in mammals to more clearly establish this relationship. Here, the authors found that HAPLN2 becomes more insoluble upon aging, specifically in the cerebellum and the extracellular matrix. The authors show that HAPLN2 can aggregate in vitro and that these aggregates can induce microglial inflammation. Moreover, chemical treatment that depletes microglial cells leads to the accumulation of aggregated HAPLN2 in younger mice. Intriguingly, injection of hyaluronidase dramatically promotes the clearance of HAPLN2 aggregates in the mouse cerebellum. This is an interesting and well-executed study that uses a broad array of methods. It is relevance to humans, especially as the authors show a similar aggregation in tissue from older individuals. While this study does always provide direct experiments to reveal some of the mechanisms at play (e.g. how are HAPLN2 cleared upon hyaluronidase treatment), I would recommend this manuscript for publication with minor revisions, which should be of great interest in our field and for PLoS biology readers.

One obvious question that is not discussed is what happens to the mice injected with hyaluronidase. Do they display any change in behavior compared to controls (e.g., better locomotion)? An extended behavioral or locomotion assessment would be beyond the scope of this study, but it would be very interesting to know (even if the results were negative).

It was unclear why the quantification of the pellet was done using a label-free approach while the soluble fraction was analyzed by TMT. This makes the direct comparison of ratios more difficult (e.g., TMT is known for compressing differences). Moreover, the 1.2-fold increase in the soluble fraction is atypical and should be better justified. Finally, it was unclear why only proteins showing an increase in both the insoluble and soluble fractions were considered. An increase in the insoluble fraction alone should suffice to draw interest, and the authors should share some information about these proteins (e.g., GO analysis).

The complete list of quantified proteins should be provided, along with their identification scores, the number of peptides and signal intensities. The raw data should also be deposited in shared platform. Some additional details should be included in the method section to better explain how the quantification was performed. For instance, was the data normalized, and were missing values imputed in the label-free quantification? Did the authors notice differences in the amount recovered from each pellet fraction?

It was unclear whether results obtained using nitrocellulose vs. PVDF membranes are expected to differ. If so, some brief information should be added to explain this. For instance, ACAN did show an increase in signal in tissue derived from older mice when using PVDF but not with the nitrocellulose membrane.

Line 236: The data in Fig. 3A show good detection in the cerebellum, but the authors cannot state that this is predominant without comparison to other brain regions.

Could the authors perform ThT or Nile Red staining of the cerebellum to determine whether HAPLN2 forms amyloid-like aggregates?

For Figures 6B and 8D, insets with enlargements of the images should be provided to better illustrate the reduction of HAPLN2 aggregation and the increase in microglial cells, which are not evident in the provided images.

Reviewer #3: Reviewer Evaluation

PBIOLOGY-D-24-03765R1

Watanabe and collaborators present an interesting study identifying new proteins prone to aggregation in the cerebellum white matter. The authors describe a protocol that relies on the isolation of sarkosyl-insoluble proteins to identify aggregate structures, which are subsequently characterized using a range of techniques, including proteomics, mass spectrometry, fluorescence, and electron microscopy. The authors eventually converge on HAPLN2, an extracellular matrix (ECM) protein that plays a key role in binding hyaluronan to other proteoglycans, primarily within the brain, where it contributes to the formation and maintenance of the ECM scaffold. HAPLN2 deficiency leads to abnormal ECM protein expression and impaired neuronal conductivity. Additionally, HAPLN2 has been shown to promote the aggregation of Parkinson's alpha-synuclein and may be present in intracellular neurofibrillary tangles (Wang et al., Front Neurosci 2019).

The authors identify HAPLN2 aggregates in both mouse and human cerebellum by immunostaining and correlate their abundance with aging. They demonstrate that pharmacological depletion of microglia increases HAPLN2 aggregation, suggesting a role for these cells in aggregate clearance. Finally, they assess the effects of HAPLN2 on microglial inflammatory activation in vivo and conclude that full-length oligomers—rather than truncated species—induce inflammation in these cells. The reviewer appreciates the extensive work completed by the authors to demonstrate the existence, and some neurological implications, of HAPLN2 aggregation. The in vivo data is compelling and shows a reasonable link between HAPLN2 and microglia in terms of clearance and inflammation. This study might eventually be of interest in the field of ageing and also to researchers working in the fields of protein aggregation and neurodegenerative diseases. However, while this work provides novel insights into HAPLN2 aggregation in aging, the origin and localization of HAPLN2 aggregates in brain tissue are not yet convincingly addressed. In my opinion, there are a number of major conceptual and experimental issues that need to be resolved before the study can be considered for publication, namely:

Major Points requiring essential revision:

1. What is the rationale for using a duplicated approach to identify insoluble proteins? The protocol for processing mouse brain tissue was very similar for both proteomics and mass spectrometry approaches, yet there is a marked discrepancy between the sarkosyl-insoluble proteins identified by these methods. What could explain this inconsistency?

2. The criteria for selecting proteins for further study are not sufficiently clear. For instance, Cystm1 had an abundance aged/young mouse ratio of 58:1 but was not examined further. The authors should clarify their selection criteria.

3. If ACAN did not form sarkosyl-insoluble aggregates, why was it identified in the proteomics assay?

4. The presence of beta-sheet conformations should be assessed using a technique other than fluorescence. Circular dichroism, a well-established method for detecting beta-sheet structures in amyloid-beta aggregates, would be more appropriate (e.g., Harada et al. Biopolymers 2011).

5. How can the authors be certain that aggregated HAPLN2 exists as isolated deposits/puncta rather than as a component of damaged myelin? Myelin declines substantially with aging in mice (Hill et al., Nat Neurosci 2018). Staining brain tissue for damaged myelin (e.g., with LFB or a similar method) alongside HAPLN2 could help assess colocalization.

6. It is surprising that no HAPLN2 aggregates were observed in other heavily myelinated brain regions, such as the corpus callosum. What might explain this?

7. Could HAPLN2 aggregates colocalize with late endosomes and lysosomes? The punctate distribution might suggest colocalization with these organelles. The authors could assess this by staining for LAMP1 alongside HAPLN2.

8. The role of hyaluronic acid in HAPLN2 aggregation remains unclear in the present study. Hyaluronic acid promoted HAPLN2 aggregate formation (Fig. 4) but did not appear necessary for aggregate maintenance (Fig. 2). However, hyaluronidase treatment in mice promoted HAPLN2 aggregate clearance in the cerebellum (Fig. 6). How do the authors reconcile these observations?

9. The authors suggest that reducing hyaluronic acid increases extracellular space, facilitating microglial access to HAPLN2 aggregates. However, this claim lacks direct evidence. The observed effect could be due solely to the reduction of hyaluronic acid, which influences aggregate formation (Fig. 2 and point 8 above).

10. Increased HAPLN2 aggregation upon microglia depletion might result from impaired myelin clearance in the absence of these cells, rather than HAPLN2 clearance per se. (see point 5). The authors might want to address that.

11. When quantifying HAPLN2 in mouse brain sections, the authors analyze individual puncta or aggregates (Figs. 3, 4, 6, 7, 9), leading to a large N. This constitutes pseudoreplication. The smallest independent unit should be at least an individual brain area field. 6-8 fields within cerebellum or hippocampus should be imaged per mouse brain section, quantified and used as N for statistical analyses.

12. In Figure 9, the 60-69-year-old donor exhibits almost no HAPLN2 aggregates, whereas the 80-89-year-old donor shows extensive aggregation. However, the quantification in panel B does not align with the presented images. The authors should address this discrepancy.

13. The Discussion section requires further clarification, specifically:

a. Lines 522-524: What is the relevance of amyloid-beta and tau in this context?

b. Why are HAPLN2 aggregates absent in the corpus callosum of aged mice?

c. Lines 546-547: Why is it significant that HAPLN2 forms aggregates via protein-protein interactions rather than complexing with hyaluronic acid?

d. Lines 554-557: The claim that microglia enhance HAPLN2 clearance due to increased extracellular space is unsubstantiated. Moreover, large aggregates are unlikely to be cleared through the glymphatic system, which primarily clears soluble peptides like Alzheimer's amyloid-beta. Is there any direct evidence demonstrating glymphatic clearance of insoluble aggregates in the literature?

e. Lines 563-565: A 0.1 pH unit decrease in brain acidity is unlikely to significantly contribute to HAPLN2 aggregation. Myelin damage may be a more plausible source.

f. Lines 572-572: Increased CD22 expression is not the sole contributor to declining microglial phagocytic capacity. The authors should explore this in greater detail.

Minor Points:

1. Paragraph beginning at line 187: Requires grammatical refinement.

2. Endotoxin-free protein production is achieved using ClearColi or similar technology. The original paper should be cited.

3. Figure legends should specify the statistical analyses used. For example, the Figure 9 legend should state: "Differences in HAPLN2 aggregate load were compared using a two-tailed Student's t-test," followed by the range of p-values and statistical significance markers.

4. Line 840 (Statistics and Reproducibility): When n represents mice, SEM should be reported instead of SD. A range of p-values and statistical significance markers should be included.

5. Line 856 (Acknowledgment section): The authors may want to include their funding sources.

6. Lines 90-92 (Introduction): PLOS Biology serves a broad audience, so biological concepts should be thoroughly explained. For example, a general reader may not be familiar with 'daf-2' and its biological significance.

7. Lines 74-76 (Introduction): Digestive exophagy is another extracellular mechanism by which microglia degrade aggregates and should be mentioned.

---

## [Editor Report · Decision Letter 2]

10 Jul 2025

Dear Dr Murata,

Thank you for your patience while we considered your revised manuscript "Aggregation of HAPLN2, a component of the perinodal extracellular matrix, is a hallmark of physiological brain aging in mice" for publication as a Research Article at PLOS Biology. This revised version of your manuscript has been evaluated by the PLOS Biology editors and the Academic Editor who is fully satisfied by your response to reviewers.

Based on our Academic Editor's assessment of your revision we are likely to accept this manuscript for publication - however before we can formally do so, we need you to address a few last data and other policy-related requests, detailed below.

**IMPORTANT: Please address the following points:

1) TITLE: We encourage you to change to the title of your paper, to reflect that HAPLN2 not only accumulates in the aging brain, but that this has functional consequences. If you agree, we suggest you change the title to something like:

"HAPLN2 forms aggregates and promotes microglial inflammation during brain aging in mice"

2) ETHICS STATEMENTS: Thank you for including an ethics statement related to the human and mouse studies performed here. We ask that you update the "Post-mortem brain samples" section of your methods to indicate whether the study was conducted according to the principles expressed in the Declaration of Helsinki. Please update the "Animals" section to include the specific national or international regulations/guidelines to which your animal care and use protocol adhered. Please note that institutional or accreditation organization guidelines (such as AAALAC) do not meet this requirement.

3) DATA: Thank you for providing your mass spec data on ProteomeXchange. I could not access this data. Can you provide me with a reviewer token so that i can check that it meets our reporting requirements? (sorry if I missed this somewhere!). Please note that we will require that this data be made public at the time your paper is published.

4) DATA: Thank you also for providing an excel file with the underlying data for your study (S1_data). Can you please add a sentence to each figure legend, pointing readers to this file?

5) CODE: Per journal policy, if you have generated any custom code during the course of this investigation, please make it available without restrictions. Please ensure that the code is sufficiently well documented and reusable, and that your Data Statement in the Editorial Manager submission system accurately describes where your code can be found.

We expect to receive your revised manuscript within two weeks.

*Published Peer Review History*

*Press*

Sincerely,

Luke

Lucas Smith, Ph.D.

Senior Editor

lsmith@plos.org

PLOS Biology

---

## [Editor Report · Decision Letter 3]

24 Jul 2025

Dear Dr Murata,

Thank you for the submission of your revised Research Article "HAPLN2 forms aggregates and promotes microglial inflammation during brain aging in mice" for publication in PLOS Biology and thank you for addressing our editorial requests in this revision. On behalf of my colleagues and the Academic Editor, Mikael Simons, I am pleased to say that we can in principle accept your manuscript for publication, provided you address any remaining formatting and reporting issues. These will be detailed in an email you should receive within 2-3 business days from our colleagues in the journal operations team; no action is required from you until then. Please note that we will not be able to formally accept your manuscript and schedule it for publication until you have completed any requested changes.

PRESS

Sincerely, 

Lucas Smith, Ph.D.

Senior Editor

PLOS Biology

lsmith@plos.org